# Bio-Inspired Image Restoration

**Yuning Cui**[1,2], **Wenqi Ren**[1,3]*, **Alois Knoll**[2]
[1]Shenzhen Campus of Sun Yat-sen University
[2]Technical University of Munich
[3]MoE Key Laboratory of Information Technology

## Abstract

Image restoration aims to recover sharp, high-quality images from degraded, low-quality inputs. Existing methods have progressively advanced from task-specific designs to general architectures, all-in-one frameworks, and composite degradation handling. Despite these advances, computational efficiency remains a critical factor for practical deployment. In this work, we present BioIR, an efficient and universal image restoration framework inspired by the human visual system. Specifically, we design two bio-inspired modules, Peripheral-to-Foveal (P2F) and Foveal-to-Peripheral (F2P), to emulate the perceptual processes of human vision, with a particular focus on the functional interplay between foveal and peripheral pathways. P2F delivers large-field contextual signals to foveal regions based on pixel-to-region affinity, while F2P propagates fine-grained spatial details through a static-to-dynamic two-stage integration strategy. Leveraging the biologically motivated design, BioIR achieves state-of-the-art performance across three representative image restoration settings: single-degradation, all-in-one, and composite degradation. Moreover, BioIR maintains high computational efficiency and fast inference speed, making it highly suitable for real-world applications. The code and pre-trained models are available at `https://github.com/c-yn/BioIR`.

## 1 Introduction

Image restoration aims to reconstruct high-quality images from their degraded counterparts [1, 2, 3, 4, 5, 6, 7, 8], which may suffer from quality loss due to adverse weather conditions or the limitations of low-end imaging devices. This task plays a critical role in supporting various downstream perceptual applications [9, 10, 11, 12], such as object detection and depth estimation. In the deep learning era, early methods have primarily focused on addressing *specific* image restoration tasks using Convolutional Neural Networks (CNNs) [13, 14, 15, 16, 17, 18] and Transformer architectures [19, 20, 21, 22, 23, 24]. Subsequently, numerous *general*-purpose methods have been proposed to tackle a variety of tasks using a shared or scalable framework, albeit with separately trained instances for each task [25, 5, 26, 27, 28, 29, 30, 31]. Recently, the concept of *all-in-one* image restoration has emerged, leading to the development of methods that incorporate degradation-aware priors to handle multiple degradations within a single model [32, 33, 34, 35, 36, 37, 38, 39, 40, 41]. The latest advancements have further extended this direction by targeting the *composite degradation* problem, which involves managing multiple degradations simultaneously [42, 43, 44, 35, 45, 46]. A detailed taxonomy of these developments is presented in Figure 1. Despite these advances, most existing algorithms remain tailored to specific scenarios and struggle to generalize effectively across diverse restoration settings.

In addition to generalization capability, computational efficiency is another critical factor for practicability. To this end, extensive efforts have been devoted to improving the efficiency of image restoration models. For instance, numerous studies have sought to enhance the computational efficiency of Transformer-based architectures for high-resolution image restoration by reducing the

---

*Corresponding author

39th Conference on Neural Information Processing Systems (NeurIPS 2025).

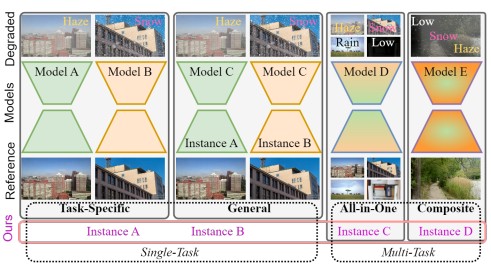 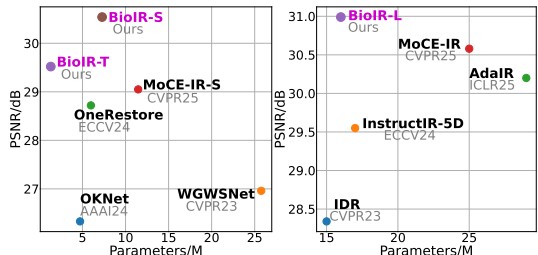

Figure 1: *Left*: Taxonomy of image restoration methods. *Middle*: Comparison of the number of parameters *vs.* PSNR on the CDD11 [42] dataset for composite degradation image restoration. *Right*: Comparison of the number of parameters *vs.* PSNR under the five-task all-in-one setting.

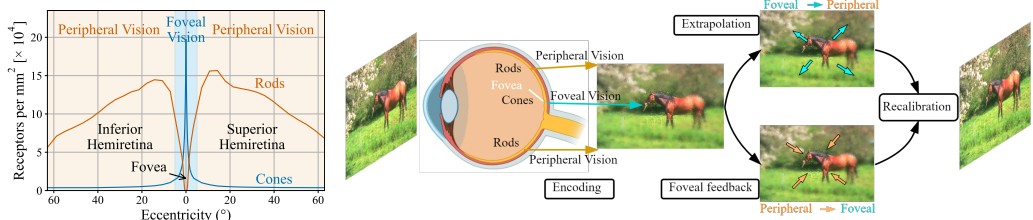

Figure 2: *Left*: Distribution of rod and cone photoreceptors across retinal eccentricity. *Right*: Conceptual overview of foveal–peripheral interactions in human vision. Due to the distinct spatial distributions and functional specializations of rods and cones, peripheral vision supports coarse scene analysis, whereas foveal vision is responsible for detailed processing. The dynamic interplay between these two systems enables the perception of a stable, coherent, and high-fidelity visual experience.

scope of self-attention operations [26, 22, 47, 27, 20, 5]. In contrast, some methods have adopted CNN-based frameworks to achieve high efficiency [30, 48, 49, 50, 51, 31, 6, 52, 53], with a subset of these further incorporating frequency-domain processing to expand the effective receptive field. More recently, a few Mamba-based image restoration algorithms have been proposed, offering an efficient mechanism for capturing long-range dependencies [54, 55, 56, 57, 58, 59, 60, 61, 62, 63, 64, 65].

Taking into account the two critical factors for practical deployment, *universality* and *efficiency*, this paper aims to develop an efficient network for universal image restoration that generalizes well across three representative settings: single-degradation, all-in-one, and composite degradation (see the left part of Figure 1(a)). To this end, we draw inspiration from the remarkable efficiency of the human visual system, whose conceptual overview is illustrated in Figure 2. As shown, the density of cone photoreceptors peaks at the fovea and declines toward the periphery [66], enabling high-acuity detail perception in a limited region. In contrast, peripheral vision covers a much larger portion of the visual field, albeit at lower resolution. After the initial encoding of visual information from both foveal and peripheral pathways, foveal signals are extrapolated to support peripheral representations [67], while peripheral object recognition benefits from integrating large-field contextual information with foveal vision [68]. This interaction is further stabilized through a recalibration process that associates foveal and peripheral inputs into a coherent percept of the visual world.

Inspired by the human perception system, we introduce two bio-inspired modules, termed foveal-to-peripheral (F2P) and peripheral-to-foveal (P2F), to facilitate the interaction between foveal and peripheral visual signals. Specifically, the P2F module delivers large-field contextual signals to local regions by leveraging pixel-to-region affinity. In contrast, the F2P module transmits fine-grained spatial details through a combination of element-wise processing and dynamic integration. The refined foveal and peripheral representations are then recalibrated via element-wise multiplication to enable high-order interactions. Building on these intuitive yet effective designs, we propose BioIR, a bio-inspired image restoration model that achieves state-of-the-art performance across three standardized settings: task-specific, all-in-one, and composite degradation, while maintaining high computational efficiency (see Figure 1). Notably, BioIR can be viewed as a higher-level general-purpose solution that relates to these three categories in much the same way as general architectures

such as Restormer [25] and NAFNet [30] relate to task-specific methods, marking a further step toward universal and practical image restoration. The main contributions are summarized as follows:

- We propose a bio-inspired, efficient, universal image restoration network that leverages two specialized modules to facilitate interaction between foveal and peripheral visual pathways.

- We introduce the foveal-to-peripheral (F2P) module, which transmits fine spatial details through a static-to-dynamic two-stage integration strategy, and the peripheral-to-foveal (P2F) module, which delivers long-range contextual information guided by pixel-to-region affinity.

- Extensive experimental results demonstrate that the proposed model achieves state-of-the-art performance across three representative image restoration settings, including four single-degradation tasks on nine datasets, two all-in-one settings, and two composite degradation benchmarks, while maintaining both high computational efficiency and fast inference speed.

## 2   Related Work

This section reviews related works from two key perspectives that support the practical application of image restoration methods, namely *versatility* and *efficiency*.

**Image Restoration.** Early approaches primarily relied on handcrafted priors to constrain the solution space [69, 70, 71]. However, designing effective priors proved to be a non-trivial challenge. Over the past decade, fueled by the rapid progress of deep learning, numerous task-specific methods based on CNNs [13, 72, 73, 18, 16, 52, 74, 75, 76] and Transformer architectures have emerged [21, 19, 22, 20, 77, 24, 78]. These methods learn generalizable priors from large-scale datasets and have demonstrated superior performance over traditional techniques on various image restoration tasks, such as deraining, dehazing, and deblurring. Despite their success, these task-specific methods often struggle to generalize to other restoration tasks due to their specialized design. To address this limitation, general-purpose architectures have gained popularity, offering strong performance across multiple tasks and sometimes even surpassing task-specific methods on their respective targets [79, 80, 29, 48, 31, 49, 30, 26, 81, 27, 82, 6, 28, 83]. For example, Restormer [25] achieves state-of-the-art results on four different tasks through an advanced Transformer architecture. However, these general architectures typically require separately trained instances for each task, forcing users to possess prior knowledge of the degradation type to select the appropriate pre-trained model.

To overcome this challenge, recent research has shifted toward all-in-one solutions that can handle multiple degradation types and levels using a single unified model [84, 41, 38, 85, 86, 87, 88]. These methods are typically trained on compound datasets aggregated from several benchmarks and often incorporate degradation-aware priors extracted from the input images. For instance, PromptIR [34] encodes priors into learnable parameters to guide the restoration process. While these models are well-suited for resource-constrained platforms, they generally assume that each input image is affected by only a single type of degradation. However, real-world scenarios often involve composite degradations, where multiple types of impairments co-occur [89, 90, 35, 91, 92, 93]. For example, night photography commonly suffers from both blurring and low-light conditions due to long exposure in dim environments [43]. Recently, CDD11 [42] was introduced as a comprehensive benchmark comprising 11 categories of degradations, including haze, rain, snow, low light, and their combinations. Correspondingly, composite degradation restoration methods have been developed, such as OneRestore [42] that leverages both textual and visual descriptors to handle these complex scenarios.

Despite these advances, existing methods largely target specific problem categories and struggle to generalize across different restoration settings [18, 25, 34, 94]. In contrast, our model is applicable to three major restoration scenarios, *i.e.,* single-degradation, all-in-one, and composite degradation, by drawing inspiration from the perceptual mechanisms of the human visual system.

**Efficient Image Restoration Networks.** Efficiency is a critical consideration for practical deployment, enabling faster processing speeds and reduced resource consumption. Recent efforts have focused on improving the computational efficiency of image restoration networks through various strategies. Specifically, several Transformer-based architectures reduce the computational overhead of self-attention by limiting its scope to smaller regions [27, 26, 22, 47, 20], such as windows or strips. In addition, some methods incorporate frequency-domain processing to efficiently model long-range

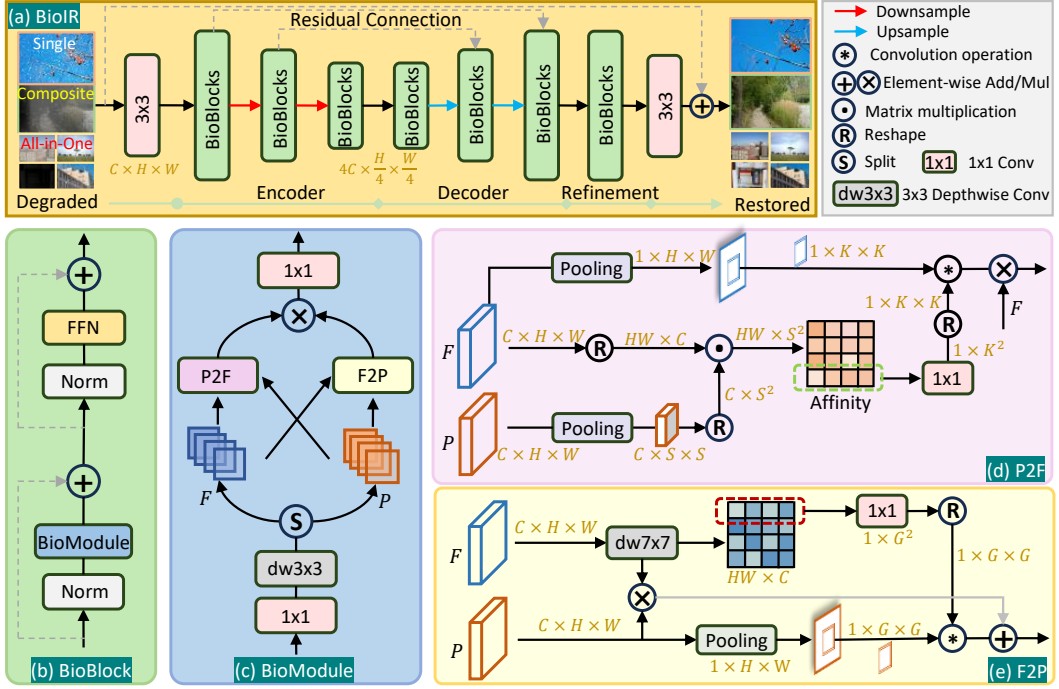

Figure 3: Architectural overview of the proposed BioIR framework for universal image restoration. (a) BioIR adopts a plain U-shaped architecture, with BioBlock (b) serving as the core building block. (c) BioModule enables dynamic interaction between two types of visual signals through the Peripheral-to-Foveal (P2F) and Foveal-to-Peripheral (F2P) modules. (d) P2F delivers large-field contextual information to local regions based on pixel-to-region affinity. (e) F2P propagates fine-grained spatial details by combining element-wise modulation and dynamic convolution.

dependencies [49, 31, 52, 51, 95]. More recently, Mamba-based frameworks have been proposed to achieve global receptive fields through advanced scanning strategies [54, 60, 56, 57, 59, 58, 55, 96].

On another front, significant progress has been made in model compression and acceleration techniques, such as quantization, pruning, and knowledge distillation [97, 98, 99, 100, 101], which further improve inference efficiency without sacrificing performance. Our proposed model is orthogonal to these techniques, providing a strong and efficient baseline that can further benefit from their integration to achieve even greater efficiency and speed.

## 3 Method

In this section, we first present an overview of the overall pipeline, highlighting the BioBlock as its core component. We then introduce the proposed BioModule, which integrates two key components: the peripheral-to-foveal (P2F) module and the foveal-to-peripheral (F2P) module.

### 3.1 Overall Pipeline

The schematic of the proposed bio-inspired network for universal image restoration is shown in Figure 3. The network adopts a plain U-shaped architecture to efficiently capture hierarchical features, without relying on advanced designs such as multi-input or multi-output strategies [31]. As illustrated, it consists of three main stages: an encoder, a decoder, and a refinement stage, along with two $3 \times 3$ convolutions used to generate feature embeddings and produce the final residual output, respectively.

More specifically, given a degraded image from a single-degradation, all-in-one, or composite degradation setting, the first $3 \times 3$ convolution embeds the 3-channel input into a feature map of size $C \times H \times W$. The encoder then progressively reduces the spatial resolution while expanding the channel dimension, resulting in a deep feature map of size $4C \times \frac{H}{4} \times \frac{W}{4}$. The decoder subsequently

restores the feature resolution back to $C \times H \times W$, with feature-level residual connections to assist the restoration process. The refinement stage further enhances the decoded feature maps [25, 29, 34]. Finally, the second $3 \times 3$ convolution produces a residual image, which is added to the degraded input via an image-level residual connection to obtain the final restored image.

The three stages are primarily composed of different numbers of BioBlocks, with architectural details illustrated in Figure 3(b). Each BioBlock follows a Transformer-style design but replaces the self-attention mechanism with the proposed BioModule, as shown in Figure 3(c). Given input features $X$, the computation within a BioBlock can be formulated as follows:

$$\hat{X} = \mathcal{F}(\mathcal{N}(\tilde{X})) + \tilde{X}, \quad \text{where} \quad \tilde{X} = \mathcal{B}(\mathcal{N}(X)) + X, \tag{1}$$

where $\hat{X}$ is the output of the BioBlock. Here, $\mathcal{N}$ denotes the normalization layer, while $\mathcal{B}$ and $\mathcal{F}$ represent the BioModule and the feed-forward network (FFN) [25], respectively.

## 3.2 Bio-Inspired Module (BioModule)

Inspired by the human visual system, which processes different types of visual signals through distinct pathways and integrates them to form a coherent perception, the proposed BioModule adopts a two-branch design that mimics this biological mechanism. Specifically, the input feature map $X \in \mathbb{R}^{C \times H \times W}$ is first refined using a $1 \times 1$ convolution followed by a $3 \times 3$ depth-wise convolution. Next, two specialized modules, P2F and F2P, are employed to extract complementary feature representations and facilitate their interaction. These features are then recalibrated via element-wise multiplication to integrate the two information streams. Finally, a $1 \times 1$ convolution produces the output feature map. The overall computation of the BioModule can be formally expressed as:

$$\hat{X} = \text{Conv}_{1\times1}(\text{P2F}(F, P) \otimes \text{F2P}(P, F)), \tag{2}$$

$$P, F = \text{Split}(\text{DWConv}_{3\times3}(\text{Conv}_{1\times1}(X))), \tag{3}$$

where *DWConv* and *Conv* denote depth-wise convolution and standard convolution, respectively, with the subscript indicating the kernel size. The *Split* operation evenly divides the feature map along the channel dimension into two parts, denoted as $P$ and $F$. Notably, the $1 \times 1$ convolution in Eq. (3) doubles the channel dimension to facilitate this splitting operation.

### 3.2.1 Peripheral-to-Foveal Module (P2F)

The P2F module delivers large-field visual signals to local spatial regions by equipping convolution operations with knowledge of long-range dependencies. To achieve this, it encodes the contextual relationships between each pixel and its surrounding regions by computing the affinity between the pixel and a set of region-representative pixels. These affinity values are then integrated through convolutional operations to dynamically generate pixel-specific convolution kernels. This enables the injection of global contextual signals into local regions via sliding-window convolutions, effectively enhancing local representations with long-range information.

The architectural details of P2F are illustrated in Figure 3(d). Given the input feature map $P \in \mathbb{R}^{C \times H \times W}$, average pooling is first applied to extract contextual representations of size $C \times S \times S$, which are then reshaped to $C \times S^2$ for affinity computation. Specifically, the affinity map $A$ is computed between these contextual features and the reshaped feature map $P$:

$$A = \mathcal{R}(F) \odot \mathcal{R}(\text{Pool}(P)) \in \mathbb{R}^{HW \times S^2}, \tag{4}$$

where $\mathcal{R}$ and *Pool* denote the reshaping and average pooling operations, respectively, and $\odot$ represents matrix multiplication. Each element in the affinity map $A$ encodes the relationship between a pixel and a specific region. Next, a $1 \times 1$ convolution is applied row-wise to $A$ to generate pixel-specific convolution kernels of size $1 \times K^2$, effectively injecting contextual information into every weight of these kernels by establishing pixel-to-region interactions.

To apply the learned kernels to $F$, we first perform average pooling on $F$ to obtain channel-compressed features of size $1 \times H \times W$, improving computational efficiency. These compressed features are then convolved with the learned kernels to produce enhanced features of size $1 \times H \times W$, which are used as attention scores to modulate $F$, resulting in the final output of the P2F module.

### 3.2.2 Foveal-to-Peripheral Module (F2P)

In contrast to P2F, which focuses on delivering global contextual information, the proposed F2P module aims to propagate fine-grained spatial details. As illustrated in Figure 3(e), F2P employs a static-to-dynamic integration strategy, element-wise modulation followed by dynamic convolution, to fully leverage the detailed local information. Specifically, given input $F \in \mathbb{R}^{C \times H \times W}$, a depth-wise convolution is first applied to extract locally refined representations. These refined features are then used to directly modulate the features $P$ through element-wise multiplication, formally expressed as:

$$\hat{X}_1 = P \otimes \tilde{F}, \quad \text{where} \ \ \tilde{F} = \text{DWConv}_{7 \times 7}(F). \tag{5}$$

In addition, $\tilde{F}$ is further utilized to generate convolutional kernels for refining $P$ in a dynamic, pixel-wise manner. Specifically, $\tilde{F}$ is passed through a $1 \times 1$ convolution to produce pixel-specific kernels. In parallel, the features $P$ are compressed to a spatial map of size $1 \times H \times W$ to enhance computational efficiency. These compressed features are then convolved with the learned kernels to integrate fine-grained information. Finally, the outputs from the element-wise modulation and the convolutional refinement stages are combined via addition, which can be formally expressed as:

$$\hat{X} = \hat{X}_1 + \hat{X}_2, \quad \text{where} \quad \hat{X}_2 = \text{Pool}(P) \circledast \text{Conv}_{1 \times 1}(\tilde{F}), \tag{6}$$

where $\circledast$ denotes the convolution operation.

## 4 Experiments and Analysis

To validate the effectiveness of the proposed model, we conduct extensive experiments across three image restoration settings: **(a)** single-degradation, **(b)** all-in-one, and **(c)** composite degradation.

- Single-degradation: The model is separately trained on nine datasets covering four restoration tasks, including image desnowing, dehazing, deraining, and low-light image enhancement.
- All-in-one: The model is evaluated under two sub-settings: three-task and five-task. It is trained on a compound dataset comprising three or five different tasks and tested on the corresponding test sets, where each image is affected by a single type of degradation.
- Composite degradation: The model is assessed on two benchmarks, LOLBlur [43] and CDD11 [42], where images suffer from up to two or three simultaneous degradation types.

We scale the model according to the complexity of the datasets to ensure a balanced trade-off between performance and efficiency. Further details on training configurations and additional results are provided in the supplementary material. In the following tables, the best and second-best results are highlighted in magenta and blue, respectively. FLOPs are calculated on a $3 \times 256 \times 256$ input patch.

### 4.1 Single-Degradation Image Restoration

**Image desnowing.** We evaluate the proposed model on three widely used datasets for image desnowing: Snow100K [15], SRRS [102], and CSD [74]. As shown in Table 1(a), our model significantly outperforms the recent Transformer-based architecture MBTF-L V2 [29] on both Snow100K and SRRS, while requiring lower computational complexity.

**Image dehazing.** We compare our tiny model with lightweight state-of-the-art algorithms on the SOTS-Indoor [103] dataset for image dehazing. As reported in Table 1(b), the proposed BioIR-T achieves a notable performance improvement of 1.05 dB in PSNR over the recent Mamba-based MaIR [59], while using only 39% of its parameters. In addition, we evaluate our base model against heavyweight dehazing methods on the Haze4K [104] dataset. Table 1(c) further demonstrates the superior performance of our model in this more challenging setting.

**Low-light image enhancement.** We further evaluate the proposed model on the LOLv2-s [105] dataset for low-light image enhancement. The quantitative results, summarized in Table 1(d), show that our tiny model outperforms the recent Mamba-based enhancer [60] by 0.36 dB in PSNR, while using fewer parameters and achieving lower computational complexity.

**Image deraining.** We first evaluate our model on the raindrop removal dataset AGAN-Data [106]. Table 1(e) shows that our model achieves a notable PSNR improvement of 0.66 dB over the recent

Table 1: Quantitative results on nine datasets across four single-degradation image restoration tasks.

(a) Snow100K, SRRS, and CSD for desnowing

| Method | Snow100K PSNR | SSIM | SRRS PSNR | SSIM | CSD PSNR | SSIM | Params (M) | FLOPs (G) |
|---|---|---|---|---|---|---|---|---|
| FocalNet[ICCV23] [28] | 33.53 | 0.95 | 31.34 | 0.98 | 37.18 | 0.99 | 3.74 | 30.6 |
| IRNeXt[ICML23] [48] | 33.61 | 0.95 | 31.91 | 0.98 | 37.29 | 0.99 | 5.46 | 42.1 |
| MBTF-L V1[ICCV23] [24] | 33.79 | 0.95 | 32.26 | 0.98 | - | - | 7.43 | 88.1 |
| ConvIR-B[PAMI24] [31] | 33.92 | 0.96 | 32.39 | 0.98 | 39.10 | 0.99 | 8.63 | 71.2 |
| MBTF-L V2[PAMI25] [29] | 34.01 | 0.96 | 32.55 | 0.98 | - | - | 7.29 | 86.0 |
| **BioIR-B**[Ours] | 34.44 | 0.96 | 32.67 | 0.98 | 39.19 | 0.99 | 7.28 | 67.81 |

(b) SOTS-Indoor for dehazing

| Method | PSNR | SSIM | Params | FLOPs |
|---|---|---|---|---|
| DeHamer[CVPR22] [19] | 36.63 | 0.988 | 132.45 | 48.93 |
| Fourmer[ICML23] [49] | 37.32 | 0.990 | 1.29 | 20.6 |
| DehazeFormer[TIP23] [20] | 38.46 | 0.994 | 4.63 | 48.64 |
| OKNet-S[AAAI24] [3] | 37.59 | 0.994 | 2.40 | 17.86 |
| DEA-Net[TIP24] [111] | 39.16 | 0.992 | 2.84 | 24.88 |
| MaIR[CVPR25] [59] | 39.45 | 0.997 | 3.40 | 24.03 |
| **BioIR-T**[Ours] | 40.50 | 0.997 | 1.32 | 16.65 |

(c) Haze4K for dehazing

| Method | PSNR | SSIM | Params | FLOPs |
|---|---|---|---|---|
| PMNet[ECCV22] [112] | 33.49 | 0.98 | 18.90 | 81.1 |
| FSNet[PAMI23] [79] | 34.12 | 0.99 | 13.29 | 110.5 |
| ConvIR-B[PAMI24] [31] | 34.15 | 0.99 | 8.63 | 71.2 |
| ConvIR-L[PAMI24] [31] | 34.50 | 0.99 | 14.83 | 129.9 |
| MBTF V1[ICCV23] [24] | 34.47 | 0.99 | 7.43 | 88.1 |
| MBTF V2[PAMI25] [29] | 34.92 | 0.99 | 7.29 | 86.0 |
| **BioIR-B**[Ours] | 35.62 | 0.99 | 7.28 | 67.81 |

(d) LOLv2-s for low-light enhancement

| Method | PSNR | SSIM | Params | FLOPs |
|---|---|---|---|---|
| MIRNet[ECCV20] [113] | 21.94 | 0.876 | 31.76 | 785 |
| Restormer[CVPR22] [25] | 21.41 | 0.830 | 26.13 | 144.25 |
| SNR-Net[CVPR22] [114] | 24.14 | 0.928 | 4.01 | 26.35 |
| Retinexformer[ICCV23] [23] | 25.67 | 0.930 | 1.61 | 15.57 |
| MambaIR[ECCV24] [54] | 25.55 | 0.929 | 4.30 | 60.66 |
| MambaLLIE[NeurIPS24] [60] | 25.87 | 0.940 | 2.28 | 20.85 |
| **BioIR-T**[Ours] | 26.22 | 0.947 | 1.32 | 16.65 |

(e) AGAN for raindrop

| Method | PSNR | SSIM |
|---|---|---|
| Restormer[CVPR22] [25] | 31.68 | 0.934 |
| IDT[PAMI22] [115] | 31.87 | 0.931 |
| MAXIM[CVPR22] [116] | 31.87 | 0.935 |
| AWRCP[ICCV23] [117] | 31.93 | 0.931 |
| FPro[ECCV24] [118] | 31.96 | 0.937 |
| AST[CVPR24] [107] | 32.32 | 0.935 |
| **BioIR-T**[Ours] | 32.98 | 0.944 |

(f) DID-Data and SPA-Data for rain streak removal

| Method | DID-Data PSNR | SSIM | SPA-Data PSNR | SSIM | Params (M) | FLOPs (G) |
|---|---|---|---|---|---|---|
| Uformer[CVPR22] [27] | 35.02 | 0.9621 | 46.13 | 0.9913 | 50.88 | 45.9 |
| Restormer[CVPR22] [25] | 35.29 | 0.9641 | 47.98 | 0.9921 | 26.13 | 144.25 |
| IDT[PAMI22] [115] | 34.89 | 0.9623 | 47.35 | 0.9930 | 16.41 | 61.9 |
| DRSformer[CVPR23] [21] | 35.35 | 0.9646 | 48.54 | 0.9924 | 33.65 | 242.9 |
| NeRD-Rain-S[CVPR24] [110] | 35.36 | 0.9647 | 48.90 | 0.9936 | 10.53 | 79.2 |
| **BioIR-B**[Ours] | 35.62 | 0.9668 | 49.39 | 0.9933 | 7.28 | 67.81 |

Table 2: Runtime on NVIDIA RTX 4090 GPU

(a) Single-degradation task

| Task/Data | Method | Time/s | PSNR |
|---|---|---|---|
| Desnowing | MBTF-L V2 | 0.558 | 34.01 |
| Snow100K | **Ours** | 0.133 | 34.44 |
| Deraining | AST | 0.389 | 32.32 |
| AGAN | **Ours** | 0.083 | 32.98 |
| Enhance | MambaLLIE | 0.093 | 25.87 |
| LOLv2-s | **Ours** | 0.038 | 26.22 |

(b) All-in-one task

| Method | Time/s | PSNR 5D | PSNR 3D |
|---|---|---|---|
| PromptIR | 0.147 | 29.15 | 32.06 |
| AdaIR | 0.144 | 30.20 | 32.69 |
| MoCE-IR | 0.112 | 30.58 | 32.73 |
| **Ours** | 0.132 | 30.99 | 32.87 |

Table 3: Quantitative results under the three-task all-in-one setting. $\sigma$ indicates the noise level.

| Method | Params | Dehazing SOTS PSNR | SSIM | Deraining Rain100L PSNR | SSIM | Denoising BSD68$_{\sigma=15}$ PSNR | SSIM | BSD68$_{\sigma=25}$ PSNR | SSIM | BSD68$_{\sigma=50}$ PSNR | SSIM | Average PSNR | SSIM |
|---|---|---|---|---|---|---|---|---|---|---|---|---|---|
| PromptIR[NeurIPS23] [34] | 36M | 30.58 | 0.974 | 36.37 | 0.972 | 33.98 | 0.933 | 31.31 | 0.888 | 28.06 | 0.799 | 32.06 | 0.913 |
| Art-PromptIR[MM24] [119] | 33M | 30.83 | 0.979 | 37.94 | 0.982 | 34.06 | 0.934 | 31.42 | 0.891 | 28.14 | 0.801 | 32.49 | 0.917 |
| UniProcessor[ECCV24] [120] | 1002M | 31.66 | 0.979 | 38.17 | 0.982 | 34.08 | 0.935 | 31.42 | 0.891 | 28.17 | 0.803 | 32.70 | 0.918 |
| AdaIR[ICLR25] [37] | 29M | 31.06 | 0.980 | 38.64 | 0.983 | 34.12 | 0.935 | 31.45 | 0.892 | 28.19 | 0.802 | 32.69 | 0.918 |
| MoCE-IR[CVPR25] [35] | 25M | 31.34 | 0.979 | 38.57 | 0.984 | 34.11 | 0.932 | 31.45 | 0.888 | 28.18 | 0.800 | 32.73 | 0.917 |
| **BioIR-L**[Ours] | 16M | 31.69 | 0.981 | 38.63 | 0.984 | 34.19 | 0.937 | 31.54 | 0.895 | 28.28 | 0.809 | 32.87 | 0.921 |

AST [107] algorithm. In addition, we conduct experiments on two benchmark datasets, DID-Data [108] and SPA-Data [109], for rain streak removal. The results, reported in Table 1(f), demonstrate that our model outperforms the specialized method NeRD-Rain-S [110], achieving a PSNR gain of 0.49 dB on the real-world SPA-Data [109] while maintaining lower computational cost.

**Inference speed comparison.** Table 2(a) presents runtime comparisons with leading algorithms on three single-degradation tasks. As shown, the proposed model achieves superior performance while maintaining faster inference speed, highlighting its practical efficiency.

## 4.2 All-in-One Image Restoration

**Three-task setting.** We begin by evaluating our model under the three-task all-in-one image restoration setting. The model is trained on a compound dataset combining three tasks: dehazing, deraining, and denoising. It is then tested on the corresponding test sets for these tasks. As shown in Table 3, our model achieves the best performance across nearly all evaluation metrics, with an average PSNR improvement of 0.14 dB over the second-best method [35]. Notably, on the BSD68 [123] dataset for image denoising, our model achieves substantial performance improvements across three different noise levels, while requiring fewer parameters than competing methods.

**Five-task setting.** Following prior works [36, 35, 37], we further evaluate our model under the five-task all-in-one setting. In addition to the three previously considered tasks, we include the GoPro [124] and LOLv1 [125] datasets for motion deblurring and low-light enhancement, respectively. As shown

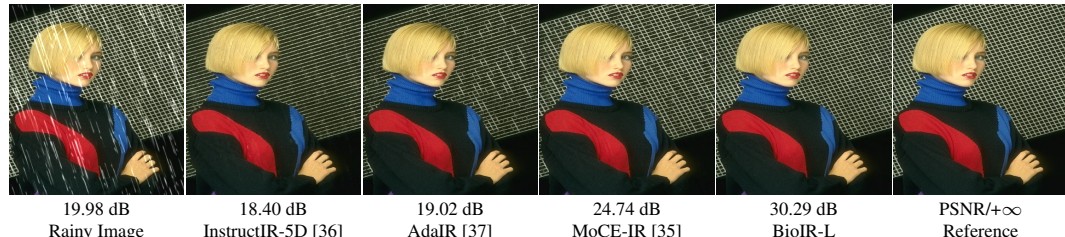

| 19.98 dB | 18.40 dB | 19.02 dB | 24.74 dB | 30.29 dB | PSNR/+∞ |
|---|---|---|---|---|---|
| Rainy Image | InstructIR-5D [36] | AdaIR [37] | MoCE-IR [35] | BioIR-L | Reference |

Figure 4: Visual results on the Rain100L [121] dataset under the five-task all-in-one setting.

Table 4: Quantitative comparisons under the five-task setting for all-in-one image restoration.

| | | Dehazing SOTS | | Deraining Rain100L | | Denoising BSD68$_{\sigma=25}$ | | Deblurring GoPro | | Low-Light LOLv1 | | Average | |
|---|---|---|---|---|---|---|---|---|---|---|---|---|---|
| Method | Params | PSNR | SSIM | PSNR | SSIM | PSNR | SSIM | PSNR | SSIM | PSNR | SSIM | PSNR | SSIM |
| IDR$_{CVPR23}$ [33] | 15M | 25.24 | 0.943 | 35.63 | 0.965 | 31.60 | 0.887 | 27.87 | 0.846 | 21.34 | 0.826 | 28.34 | 0.893 |
| PromptIR$_{NeurIPS23}$ [34] | 36M | 26.54 | 0.949 | 36.37 | 0.970 | 31.47 | 0.886 | 28.71 | 0.881 | 22.68 | 0.832 | 29.15 | 0.904 |
| Gridformer$_{IJCV24}$ [122] | 34M | 26.79 | 0.951 | 36.61 | 0.971 | 31.45 | 0.885 | 29.22 | 0.884 | 22.59 | 0.831 | 29.33 | 0.904 |
| InstructIR-5D$_{ECCV24}$ [36] | 17M | 27.10 | 0.956 | 36.84 | 0.973 | 31.40 | 0.873 | 29.40 | 0.886 | 23.00 | 0.836 | 29.55 | 0.908 |
| AdaIR$_{ICLR25}$ [37] | 29M | 30.53 | 0.978 | 38.02 | 0.981 | 31.35 | 0.889 | 28.12 | 0.858 | 23.00 | 0.845 | 30.20 | 0.910 |
| MoCE-IR$_{CVPR25}$ [35] | 25M | 30.48 | 0.974 | 38.04 | 0.982 | 31.34 | 0.887 | 30.05 | 0.899 | 23.00 | 0.852 | 30.58 | 0.919 |
| **BioIR-L$_{Ours}$** | 16M | 31.77 | 0.981 | 38.75 | 0.985 | 31.52 | 0.894 | 29.61 | 0.889 | 23.29 | 0.862 | 30.99 | 0.922 |

Table 5: PSNR results of directly applying the model pre-trained under the five-task all-in-one setting to three denoising datasets: BSD68 [123], Urban100 [127], and Kodak24 [128].

| | BSD68 [123] | | | Urban100 [127] | | | Kodak24 [128] | | | |
|---|---|---|---|---|---|---|---|---|---|---|
| Method | $\sigma=15$ | $\sigma=25$ | $\sigma=50$ | $\sigma=15$ | $\sigma=25$ | $\sigma=50$ | $\sigma=15$ | $\sigma=25$ | $\sigma=50$ | Average |
| TransWeather$_{CVPR22}$ [129] | 31.16 | 29.00 | 26.08 | 29.64 | 27.97 | 26.08 | 31.67 | 29.64 | 26.74 | 28.66 |
| IDR$_{CVPR23}$ [33] | 34.11 | 31.60 | 28.14 | 33.82 | 31.29 | 28.07 | 34.78 | 32.42 | 29.13 | 31.48 |
| InstructIR-5D$_{ECCV24}$ [36] | 34.00 | 31.40 | 28.15 | 33.77 | 31.40 | 28.13 | 34.70 | 32.26 | 29.16 | 31.44 |
| MoCE-IR$_{CVPR25}$ [35] | 34.00 | 31.34 | 28.07 | 34.01 | 31.59 | 28.20 | 34.87 | 32.38 | 29.20 | 31.52 |
| AdaIR$_{ICLR25}$ [37] | 34.01 | 31.35 | 28.06 | 34.10 | 31.68 | 28.29 | 34.89 | 32.38 | 29.21 | 31.55 |
| **BioIR-L$_{Ours}$** | 34.18 | 31.52 | 28.27 | 34.50 | 32.16 | 28.92 | 35.15 | 32.67 | 29.55 | 31.88 |

in Table 4, our model achieves the best results in most categories. Notably, it establishes a new state-of-the-art by outperforming the second-best algorithm, MoCE-IR [35], with an average PSNR improvement of 0.41 dB across all datasets. While MoCE-IR achieves higher scores on the GoPro dataset, this is primarily due to its reliance on a specialized motion deblurring backbone [126]. Additionally, our model surpasses the recent frequency-based AdaIR [37] by a notable margin of 0.79 dB in PSNR. Importantly, these improvements are achieved with fewer parameters, highlighting the efficiency and effectiveness of our bio-inspired design. Visual results in Figure 4 further demonstrate the superiority of our model, showing that its output is significantly closer to the reference image, whereas the first two methods produce results that perform even worse than the degraded rainy input.

Additionally, we assess the generalization capability of our model by directly applying it to unseen denoising datasets, including Urban100 [127] and Kodak24 [128]. As reported in Table 5, our model consistently outperforms recent strong competitors across all evaluation metrics. In particular, it achieves a notable average PSNR improvement of 0.34 dB over the heavyweight AdaIR [37], further demonstrating its robustness and generalization ability.

**Runtime comparison.** Table 2(b) reports the inference time of state-of-the-art all-in-one algorithms on the Rain100L [121] dataset. As shown, our model achieves significantly higher accuracy than the competing methods under both settings, while maintaining a comparable inference speed.

*Discussion.* Despite not relying on any explicit degradation-related priors, as employed in previous methods [34, 32, 37], our model achieves strong performance in both all-in-one settings. We attribute this success to two key factors: (*i*) the large-field signals extracted by the peripheral vision branch, which capture the contextual information of degraded images and can serve as implicit priors to guide the restoration of local details in each module; and (*ii*) the bio-inspired architecture, which endows the model with powerful representation learning capabilities, enabling it to recover sharp features from degraded inputs without relying on explicit, task-specific priors.

Table 6: Quantitative evaluation on the LOLBlur [43] dataset for composite degradation.

| Method | MIMO [1] | NAFNet [30] | LEDNet [43] | Restormer [25] | RetinexFormer [23] | DarkIR-M [44] | DarkIR-L [44] | **BioIR-T** |
|---|---|---|---|---|---|---|---|---|
| Venue | ICCV21 | ECCV22 | ECCV22 | CVPR22 | ICCV23 | CVPR25 | CVPR25 | Ours |
| PSNR | 22.41 | 25.36 | 25.74 | 26.72 | 26.02 | 27.00 | 27.30 | 27.70 |
| SSIM | 0.835 | 0.882 | 0.850 | 0.902 | 0.887 | 0.883 | 0.898 | 0.908 |
| Params (M) | 6.8 | 12.05 | 7.4 | 26.13 | 1.61 | 3.31 | 12.96 | 1.32 |

Table 7: Quantitative comparison for composite degradation image restoration on the 11 degradation categories of the CDD11 [42] dataset. Results are reported in terms of PSNR and SSIM .

| Method | Params | Low (L) | Haze (H) | Rain (R) | Snow (S) | L+H | L+R | L+S | H+R | H+S | L+H+R | L+H+S | Average |
|---|---|---|---|---|---|---|---|---|---|---|---|---|---|
| AirNet | 9M | 24.83 .778 | 24.21 .951 | 26.55 .891 | 26.79 .919 | 23.23 .779 | 22.82 .710 | 23.29 .723 | 22.21 .868 | 23.29 .901 | 21.80 .708 | 22.24 .725 | 23.75 .814 |
| PromptIR | 36M | 26.32 .805 | 26.10 .969 | 31.56 .946 | 31.53 .960 | 24.49 .789 | 25.05 .771 | 24.51 .761 | 24.54 .924 | 23.70 .925 | 23.74 .752 | 23.33 .747 | 25.90 .850 |
| WGWSNet | 26M | 24.39 .774 | 27.90 .982 | 33.15 .964 | 34.43 .973 | 24.27 .800 | 25.06 .772 | 24.60 .765 | 27.23 .955 | 27.65 .960 | 23.90 .772 | 23.97 .771 | 26.96 .863 |
| WeatherDiff | 83M | 23.58 .763 | 21.99 .904 | 24.85 .885 | 24.80 .888 | 21.83 .756 | 22.69 .730 | 22.12 .707 | 21.25 .868 | 21.99 .868 | 21.23 .716 | 21.04 .698 | 22.49 .799 |
| OneRestore | 6M | 26.48 .826 | 32.52 .990 | 33.40 .964 | 34.31 .973 | 25.79 .822 | 25.58 .799 | 25.19 .789 | 29.99 .957 | 30.21 .964 | 24.78 .788 | 24.90 .791 | 28.47 .878 |
| MoCE-IR-S | 11M | 27.26 .824 | 32.66 .990 | 34.31 .970 | 35.91 .980 | 26.24 .817 | 26.25 .800 | 26.04 .793 | 29.93 .964 | 30.19 .970 | 25.41 .789 | 25.39 .790 | 29.05 .881 |
| **BioIR-T** | 1M | 27.14 .834 | 34.77 .992 | 34.48 .970 | 36.26 .979 | 26.11 .830 | 26.17 .809 | 26.01 .804 | 31.64 .968 | 31.60 .972 | 25.22 .799 | 25.34 .802 | 29.52 .887 |
| **BioIR-B** | 7M | 27.52 .838 | 36.95 .995 | 35.35 .974 | 37.28 .981 | 26.79 .835 | 26.78 .817 | 26.63 .812 | 32.77 .973 | 33.39 .977 | 26.20 .811 | 26.28 .811 | 30.54 .893 |

Table 8: Ablation results for the proposed bio-inspired components and alternative design choices.

| (a) Ablation studies for individual components. | | | | | | (b) Peripheral vision scope ($S^2$, Fig. 3d). | | | | (c) Foveal vision scope (dw, Fig. 3e). | | | |
|---|---|---|---|---|---|---|---|---|---|---|---|---|---|
| Recalibration | P2F | F2P | PSNR | FLOPs | Params | Scope | PSNR | FLOPs | Params | Kernel | PSNR | FLOPs | Params |
| | | | 33.18 | 15.15 | 1.31 | $1 \times 1$ | 36.10 | 14.37 | 1.23 | $1 \times 1$ | 34.49 | 14.53 | 1.24 |
| ✓ | | | 33.67 | 14.34 | 1.23 | $2 \times 2$ | 36.27 | 14.38 | 1.23 | $3 \times 3$ | 35.32 | 14.66 | 1.24 |
| | ✓ | | 36.64 | 15.38 | 1.25 | $4 \times 4$ | 36.20 | 14.43 | 1.23 | $5 \times 5$ | 36.10 | 14.93 | 1.26 |
| | | ✓ | 36.96 | 15.61 | 1.29 | $8 \times 8$ | 36.45 | 14.62 | 1.23 | $7 \times 7$ | 36.59 | 15.33 | 1.28 |
| ✓ | ✓ | ✓ | 38.26 | 16.65 | 1.32 | $16 \times 16$ | 36.64 | 15.38 | 1.25 | $9 \times 9$ | 36.83 | 15.87 | 1.30 |

## 4.3 Composite Degradation Image Restoration

**Two-degradation setting.** We first evaluate our model on the LOLBlur [43] dataset, which presents a composite degradation scenario involving both low-light conditions and blur. Table 6 shows that our model significantly outperforms the recent specialized method DarkIR-L [44] by 0.4 dB, while requiring only 10% of its parameters. Compared to DarkIR-M, our model achieves an even larger improvement of up to 0.7 dB in PSNR, while using less than half the number of parameters.

**Three-degradation setting.** We further compare our model against state-of-the-art multi-task image restoration methods on the more challenging CDD11 [42] dataset, which involves composite degradations across 11 degradation categories, with each image affected by up to three simultaneous degradation types. As reported in Table 7, our base model, BioIR-B, consistently achieves the best performance across all evaluation metrics, significantly outperforming the second-best method, MoCE-IR-S, by an average margin of 1.02 dB in PSNR. Notably, our tiny model, BioIR-T, surpasses most competing methods on the majority of metrics while using only approximately 1M parameters, highlighting the efficiency of our design.

## 4.4 Ablation Study

We conduct ablation studies by training the dehazing model on the RESIDE-Indoor [103] dataset for 100k iterations and evaluating its performance on the corresponding SOTS-Indoor [103] test set. More ablation study results are provided in the supplementary material.

**Effects of individual components.** Table 8(a) presents the results of adding the proposed components to a baseline model, which consists of only two $1 \times 1$ convolutions and a $3 \times 3$ depth-wise convolution within the BioModule. This baseline achieves 33.18 dB on the SOTS-Indoor [103] dataset. Introducing feature splitting and the recalibration mechanism via element-wise multiplication improves the performance by 0.49 dB, while slightly reducing computational overhead. Adding the two key components, P2F and F2P, boosts performance by 3.46 dB and 3.78 dB over the baseline, respectively, demonstrating the effectiveness of the proposed bio-inspired design. Finally, applying recalibration to the outputs of both P2F and F2P yields the best result, achieving a 5.08 dB PSNR improvement over the baseline with only a negligible increase in computational cost.

**Peripheral vision scope.** The peripheral vision branch in P2F provides large-field contextual signals to support foveal processing. We investigate the effect of varying the peripheral vision scope, which is controlled by the spatial resolution $S^2$ of the pooled feature map (Figure 3d). A larger $S$ corresponds to a smaller peripheral field. As shown in Table 8(b), increasing $S$ generally improves performance, suggesting that a more focused peripheral scope benefits the model. In our final design, we adopt a resolution of $16 \times 16$, which provides a favorable trade-off between performance and efficiency.

**Foveal vision scope.** The foveal vision branch in F2P delivers fine-grained spatial information to the peripheral pathway. As shown in Table 8(c), expanding the spatial extent of foveal processing consistently improves performance, but at the expense of increased computational cost. This observation aligns with the biological organization of the human visual system, where the separation and cooperation of foveal and peripheral pathways are essential for achieving both stability and efficiency in perception. Based on this trade-off, we adopt a kernel size of $7 \times 7$ in our final model to achieve a balanced compromise between accuracy and computational efficiency.

## 5 Conclusion

This study presents an efficient and universal image restoration network inspired by the human visual system. Specifically, the proposed peripheral-to-foveal (P2F) module extracts large-field contextual signals and integrates them into local regions to guide the restoration process, while the foveal-to-peripheral (F2P) module propagates fine-grained local features to enhance peripheral representations. A recalibration mechanism further harmonizes these two complementary information streams. Built upon this simple yet highly effective bio-inspired design, the proposed BioIR network achieves state-of-the-art performance across three major categories of image restoration tasks, including nine datasets for four single-degradation tasks, two all-in-one task settings, and two composite degradation benchmarks, while maintaining high computational efficiency and fast inference speed.

## Acknowledgement

This work was supported by the National Natural Science Foundation of China under Grants 62311530686, 62322216, U24B20175, the Guangdong Basic and Applied Basic Research Foundation under Grant 2023A1515012839, and the Fundamental Research Funds for the Central Universities, Sun Yat-sen University under Grant 23lgbj015.

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

# A Technical Appendices and Supplementary Material

## A.1 Implementation Details

This subsection presents the implementation details for the three image restoration settings. To accommodate the varying complexities of different datasets and scenarios, we provide three architectural variants of our model: BioIR-T (Tiny), BioIR-B (Base), and BioIR-L (Large). As summarized in Table 9, these variants are scaled by adjusting the number of channels and the number of BioBlocks at each resolution level, while keeping all other architectural components unchanged across versions.

**Single-degradation/composite degradation image restoration.** Following previous methods [31, 1, 28], the proposed model is trained using the $l_1$ loss computed in both the spatial and frequency domains, optimized with the Adam optimizer. The initial learning rate is set to $1 \times 10^{-3}$ and is progressively reduced to $1 \times 10^{-7}$ using a cosine annealing schedule. Consistent with prior Transformer-based approaches [25], our models are generally trained for 300K iterations. For deraining tasks, evaluation is performed on the Y channel in the YCbCr color space, following standard protocols [107, 21, 110]. All experiments are conducted using PyTorch on NVIDIA Tesla A100 80G GPUs.

Table 9: The configurations of different BioIR variants.

| BioIR | Channels | BioBlocks | Parameters | FLOPs |
|---|---|---|---|---|
| Tiny (T) | [32, 64, 128, 128, 64, 32, 32] | [1, 1, 2, 2, 1, 1, 4] | 1.32M | 16.65G |
| Base (B) | [48, 96, 192, 192, 96, 48, 48] | [3, 3, 6, 6, 3, 3, 4] | 7.28M | 67.81G |
| Large (L) | [48, 96, 192, 192, 96, 48, 48] | [6, 6, 14, 14, 6, 6, 4] | 15.85M | 125.64G |

**All-in-one image restoration.** The datasets used in the two all-in-one image restoration settings are summarized in Table 10. The three-task setting includes dehazing, deraining, and denoising, while the five-task setting additionally incorporates motion deblurring and low-light image enhancement for both training and evaluation. Noisy images are synthesized by adding Gaussian noise with $\sigma \in \{15, 25, 50\}$ to clean images. Our dataset configurations closely follow those adopted in previous all-in-one studies [34, 35, 37, 36].

Following [35], the proposed model is trained using dual-domain $l_1$ loss functions on $128 \times 128$ image patches. The batch size is set to 32, and the model is trained for 100 epochs in the three-task setting and 150 epochs in the five-task setting. The initial learning rate is set to $2 \times 10^{-4}$. For data augmentation, random horizontal and vertical flips are applied.

Table 10: Datasets used in the experiments for two all-in-one settings.

| Task | Dehazing | Deraining | Denoising | Deblurring | Enhancement |
|---|---|---|---|---|---|
| Training set | RESIDE$-\beta$ [103] | Rain100L [121] | BSD400 [130], WED [131] | GoPro [124] | LOLv1 [125] |
| Test set | SOTS-Outdoor [103] | Rain100L [121] | BSD68 [123], Urban100 [127], Kodak24 [128] | GoPro [124] | LOLv1 [125] |

## A.2 Additional ablation studies

Table 11: Additional ablation studies for F2P. The kernel size of the depthwise convolution in F2P is set to $7 \times 7$.

| Method | PSNR | FLOPs | Params |
|---|---|---|---|
| F2P (one-stage, G=3 in Fig. 3(e)) | 35.85 | 15.33 | 1.28 |
| F2P (two-stage, G=3) | 36.59 | 15.33 | 1.28 |
| F2P (two-stage, G=5) | 36.96 | 15.61 | 1.29 |
| F2P (two-stage, G=7) | 37.08 | 16.02 | 1.31 |

**More ablations for F2P.** We first conduct experiments to validate the effectiveness of the two-stage design in the F2P module. As reported in Table 11, the one-stage variant, obtained by removing the element-wise multiplication, achieves a PSNR of 35.85 dB, while the proposed two-stage design improves performance to 36.59 dB without introducing additional computational overhead. We

further investigate the impact of the kernel size ($G$ in Fig. 3(e)) used in the dynamic convolution. The results show that performance consistently improves as the kernel size increases. Based on this trade-off between computational cost and accuracy, we select $G = 5$ in our final model.

**More ablations for P2F.** Similarly, we investigate the impact of increasing the kernel size ($K$ in Fig. 3(d)) of the dynamic convolution in the P2F module. The results, summarized in Table 12, show performance trends across different kernel sizes. To maintain a lightweight design, we adopt $K = 3$ in our final model.

Table 12: Additional ablation studies for P2F. $S$ is set to 16 in P2F.

| Method | PSNR | FLOPs | Params |
|---|---|---|---|
| P2F (K=3 in Fig. 3(d)) | 36.64 | 15.38 | 1.25 |
| P2F (K=5) | 36.8 | 17.2 | 1.30 |

## A.3  Evaluation on perceptual metrics

In addition to PSNR and SSIM [132], this subsection reports evaluation results using the perceptual metric LPIPS [133] to further assess the effectiveness of the proposed model. Table 13 compares our model with leading all-in-one algorithms. As shown, our model achieves the best LPIPS scores on most datasets, demonstrating superior perceptual quality compared to competing methods.

Table 13: Comparisons under the five-task (top) and three-task (bottom) all-in-one image restoration settings, evaluated in terms of LPIPS [133] (lower is better).

| Method | *Dehazing* SOTS | *Deraining* Rain100L | *Denoising* (BSD68) $\sigma = 15$ | $\sigma = 25$ | $\sigma = 50$ | *Deblurring* GoPro | *Low-Light* LOLv1 |
|---|---|---|---|---|---|---|---|
| AdaIR$_{ICLR25}$ [37] | 0.0129 | 0.0139 | 0.0634 | 0.1114 | 0.2105 | 0.1902 | 0.1206 |
| MoCE-IR$_{CVPR25}$ [35] | 0.0126 | 0.0137 | 0.0610 | 0.1029 | 0.1945 | 0.1444 | 0.1183 |
| **BioIR-L**$_{Ours}$ | 0.0107 | 0.0107 | 0.0569 | 0.1015 | 0.1934 | 0.1450 | 0.1122 |
| AdaIR$_{ICLR25}$ [37] | 0.0116 | 0.0118 | 0.0611 | 0.1079 | 0.2112 | - | - |
| MoCE-IR$_{CVPR25}$ [35] | 0.0119 | 0.0106 | 0.0575 | 0.1008 | 0.1906 | - | - |
| **BioIR-L**$_{Ours}$ | 0.0110 | 0.0108 | 0.0509 | 0.0924 | 0.1859 | - | - |

## A.4  Additional visual results

This subsection presents additional qualitative results for single-degradation, all-in-one, and composite degradation image restoration tasks, organized as follows:

- Single-degradation: Figure 5 (SRRS [102], desnowing), Figure 6 (Haze4k [104], dehazing), Figure 7 (LOLv2-s [105], enhancement)
- Composite degradation: Figure 8 (LOLBlur [43]), Figure 9 (CDD11 [42]), Figure 10 (Real-LOLBlur [43])
- All-in-one: Figures 11, 12 (Rain100L [121], BSD68 [123], five-task setting)

These results further demonstrate the superior capability of our model in addressing various types of degradations across diverse image restoration scenarios.

## A.5  Limitation and Broader Impact

This work presents a bio-inspired image restoration network that achieves state-of-the-art performance across three representative restoration settings, while maintaining high computational efficiency. However, the human visual perception process is highly complex and remains an active area of investigation [68, 134]. The bio-inspired mechanisms proposed in this study represent only a preliminary attempt to mimic this process. Additionally, the current recalibration strategy is implemented using simple element-wise multiplication. Designing more advanced recalibration mechanisms may further improve performance and robustness, which we consider a promising direction for future work.

While this study focuses on the academic advancement of universal image restoration, the proposed algorithm holds potential for various practical applications. It may bring positive social impacts, such as enhancing image quality captured under adverse conditions or with low-end devices. However, it also raises potential concerns, including the risk of privacy leakage when enhancing sensitive content. Nonetheless, we believe the benefits significantly outweigh the risks, and users are encouraged to apply additional privacy-preserving techniques to safeguard sensitive information in images.

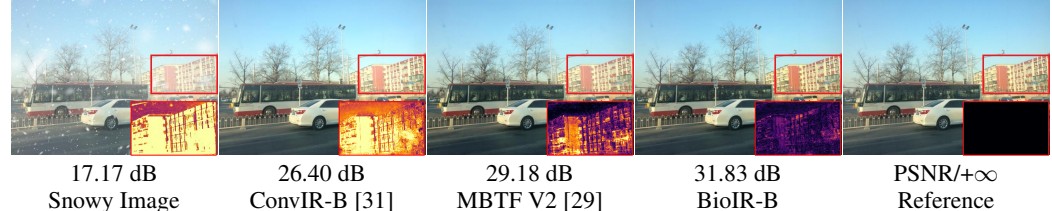

| 17.17 dB | 26.40 dB | 29.18 dB | 31.83 dB | PSNR/+∞ |
| Snowy Image | ConvIR-B [31] | MBTF V2 [29] | BioIR-B | Reference |

Figure 5: Visual results on the SRRS [102] dataset for image desnowing. The bottom-right sub-images present the $L_1$ error maps computed between the restored and reference regions enclosed by small red boxes, where brighter pixels indicate larger errors, highlighting the superiority of our approach.

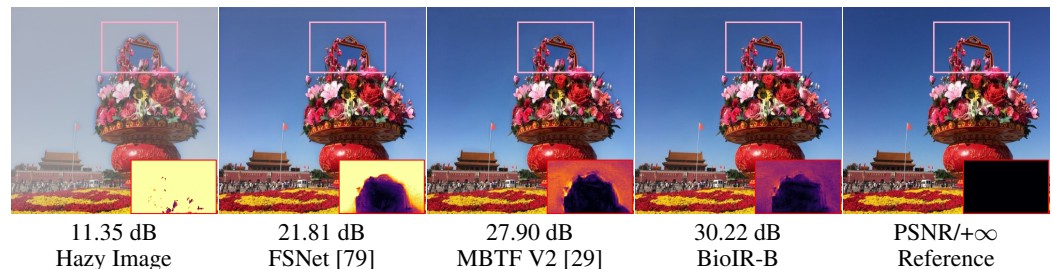

| 11.35 dB | 21.81 dB | 27.90 dB | 30.22 dB | PSNR/+∞ |
| Hazy Image | FSNet [79] | MBTF V2 [29] | BioIR-B | Reference |

Figure 6: Visual results on the Haze4k [104] dataset for image dehazing. The bottom-right sub-images present the $L_1$ error maps computed between the restored and reference regions enclosed by small pink boxes, where brighter pixels indicate larger errors, highlighting the superiority of our approach.

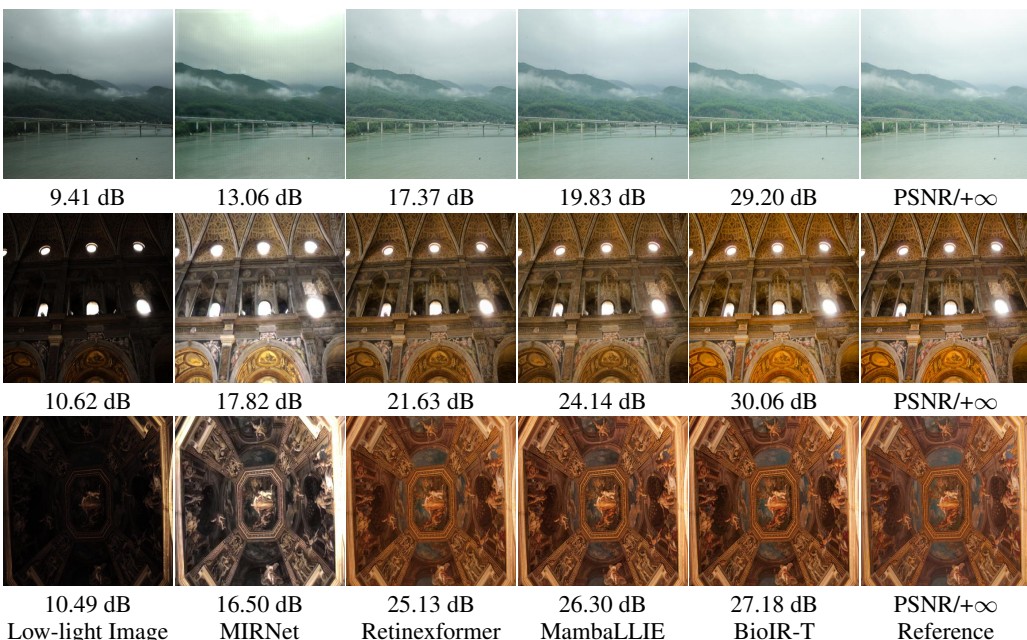

| 9.41 dB | 13.06 dB | 17.37 dB | 19.83 dB | 29.20 dB | PSNR/+∞ |
| 10.62 dB | 17.82 dB | 21.63 dB | 24.14 dB | 30.06 dB | PSNR/+∞ |
| 10.49 dB | 16.50 dB | 25.13 dB | 26.30 dB | 27.18 dB | PSNR/+∞ |
| Low-light Image | MIRNet | Retinexformer | MambaLLIE | BioIR-T | Reference |

Figure 7: Visual comparison on the LOLv2-s [105] dataset for low-light image enhancement.

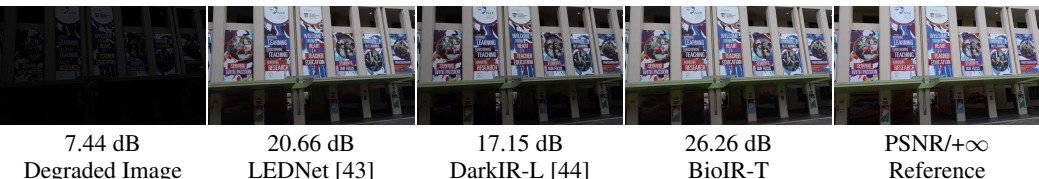

| 7.44 dB | 20.66 dB | 17.15 dB | 26.26 dB | PSNR/+∞ |
| Degraded Image | LEDNet [43] | DarkIR-L [44] | BioIR-T | Reference |

Figure 8: Qualitative results on the LOLBlur [43] dataset under composite degradations. Our model achieves more effective brightness restoration and detail recovery from low-light and blurred inputs.

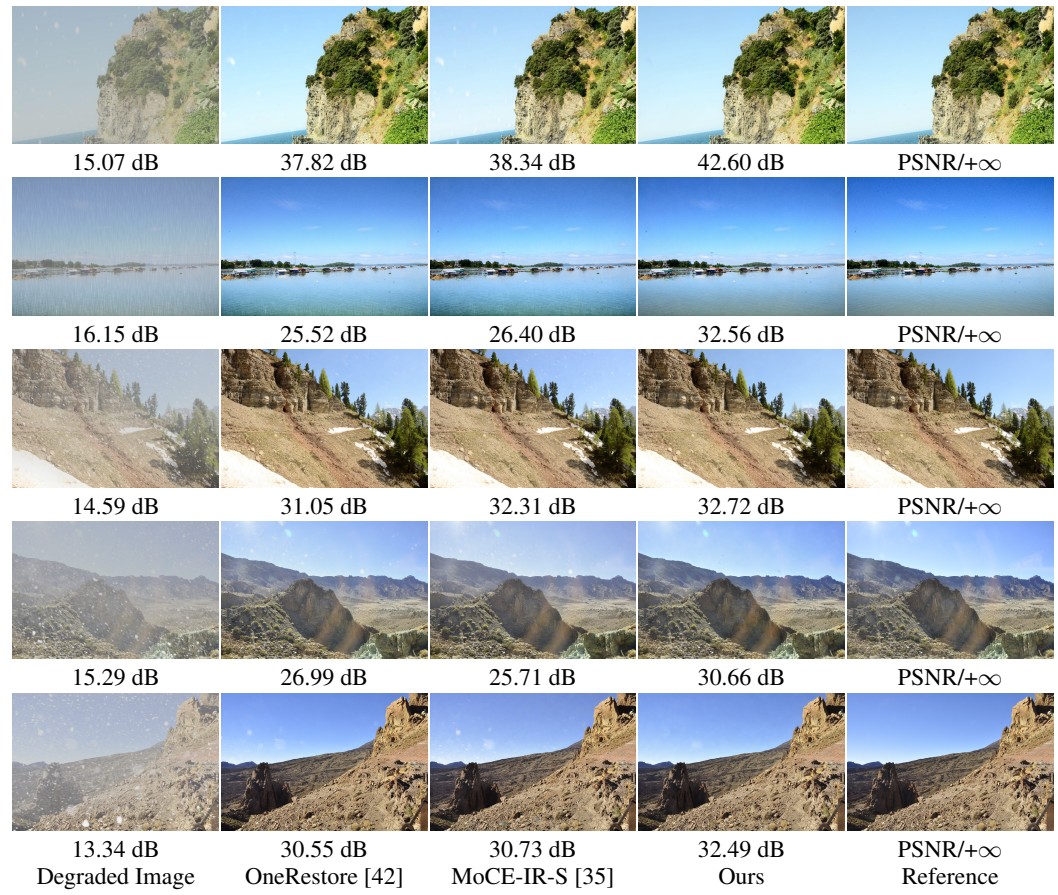

| 15.07 dB | 37.82 dB | 38.34 dB | 42.60 dB | PSNR/+∞ |
| 16.15 dB | 25.52 dB | 26.40 dB | 32.56 dB | PSNR/+∞ |
| 14.59 dB | 31.05 dB | 32.31 dB | 32.72 dB | PSNR/+∞ |
| 15.29 dB | 26.99 dB | 25.71 dB | 30.66 dB | PSNR/+∞ |
| 13.34 dB | 30.55 dB | 30.73 dB | 32.49 dB | PSNR/+∞ |
| Degraded Image | OneRestore [42] | MoCE-IR-S [35] | Ours | Reference |

Figure 9: Visual results on the CDD11 [42] dataset for composite degradations.

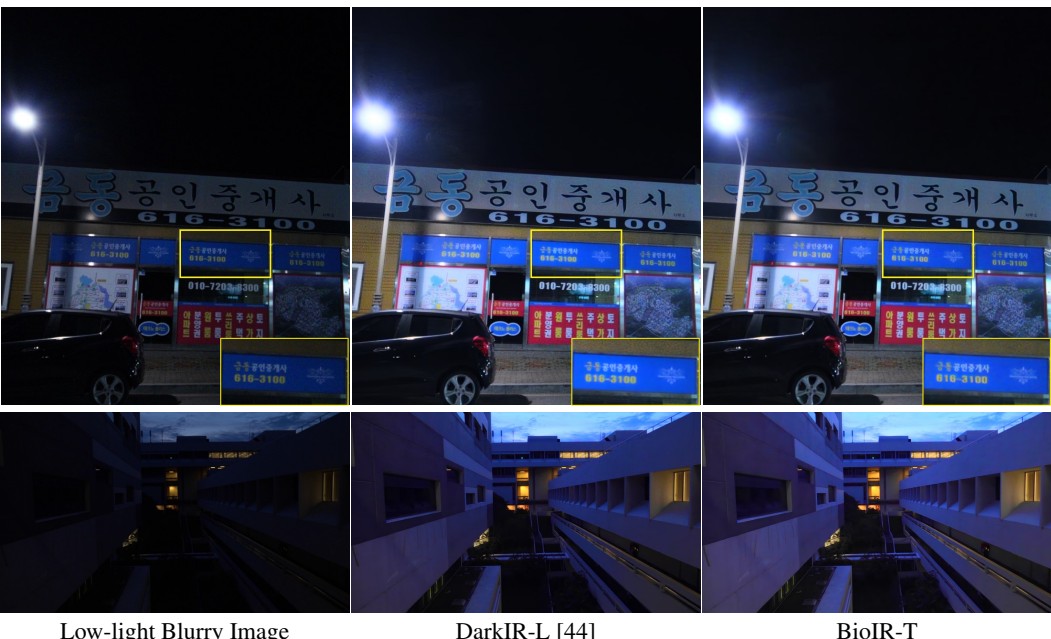

| Low-light Blurry Image | DarkIR-L [44] | BioIR-T |

Figure 10: Generalization evaluation by directly applying the model pretrained on LOLBlur [43] to real-world night blurry images. The top example is from the RealBlur [90] dataset, while the bottom example is captured in an uncontrolled real-world setting. Our model more effectively restores both text details and overall brightness.

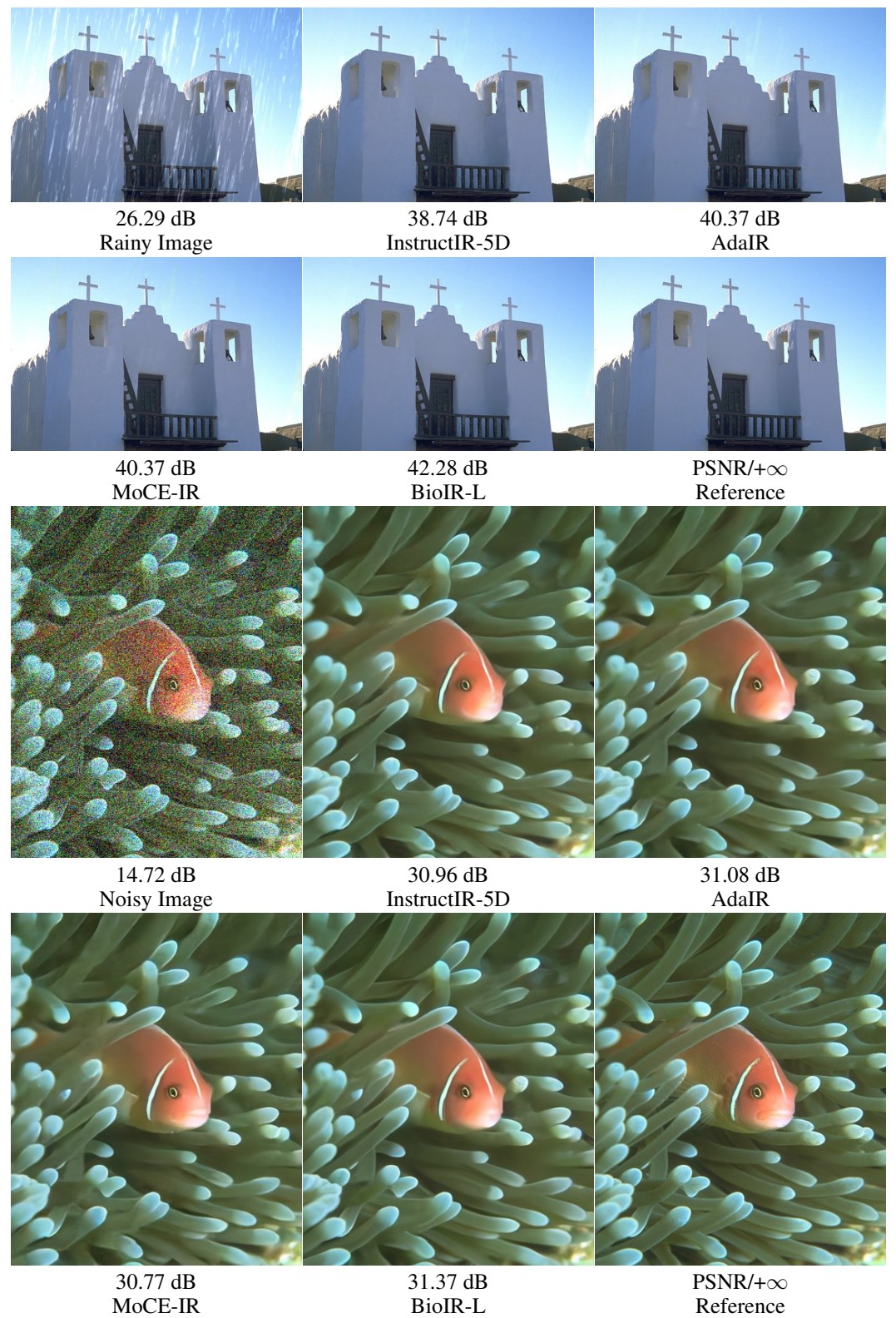

| 26.29 dB | 38.74 dB | 40.37 dB |
| Rainy Image | InstructIR-5D | AdaIR |

| 40.37 dB | 42.28 dB | PSNR/+∞ |
| MoCE-IR | BioIR-L | Reference |

| 14.72 dB | 30.96 dB | 31.08 dB |
| Noisy Image | InstructIR-5D | AdaIR |

| 30.77 dB | 31.37 dB | PSNR/+∞ |
| MoCE-IR | BioIR-L | Reference |

Figure 11: Visual results under the five-task all-in-one setting.

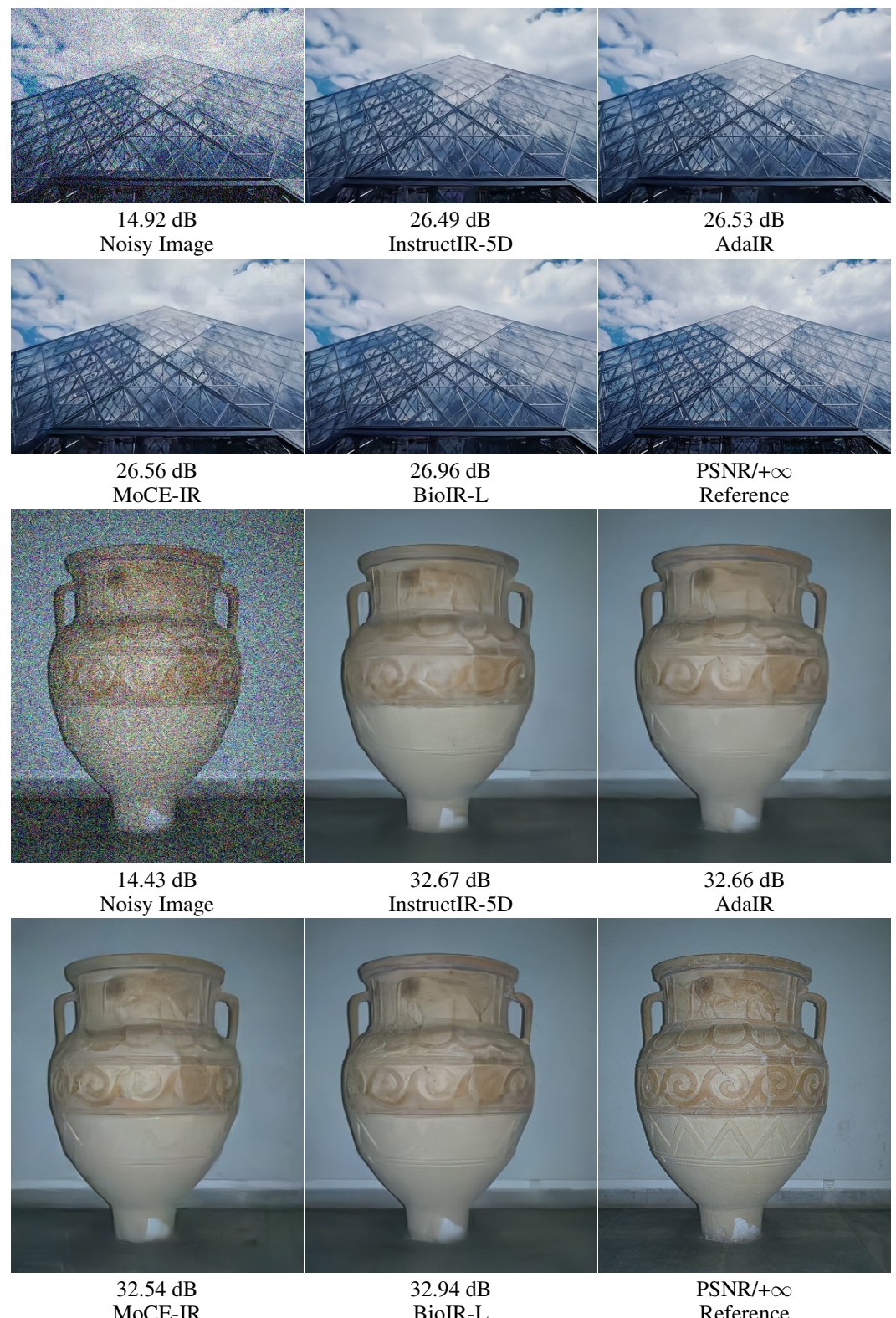

| 14.92 dB | 26.49 dB | 26.53 dB |
| Noisy Image | InstructIR-5D | AdaIR |

| 26.56 dB | 26.96 dB | PSNR/+∞ |
| MoCE-IR | BioIR-L | Reference |

| 14.43 dB | 32.67 dB | 32.66 dB |
| Noisy Image | InstructIR-5D | AdaIR |

| 32.54 dB | 32.94 dB | PSNR/+∞ |
| MoCE-IR | BioIR-L | Reference |

Figure 12: Visual results under the five-task all-in-one setting.

