# OpenReview forum: "Bio-Inspired Image Restoration"
_NeurIPS.cc/2025/Conference — NeurIPS 2025 poster_

### Official Review · Reviewer_jF93 · 2025-06-21

**Clarity:** 2
**Significance:** 3
**Originality:** 3
**Rating:** 4
**Confidence:** 4

**Summary:**

The paper proposes BioIR, aimed to build a generalizable and efficient model for image restoration. Specifically, BioIR is a U-shaped network that replaces self-attention with two biologically-inspired operators: Peripheral-to-Foveal (P2F), which routes large-field context to local features via pixel-region affinities, and Foveal-to-Peripheral (F2P), which sends fine spatial details back to the peripheral stream through a static-to-dynamic integration. BioIR reaches state-of-the-art PSNR/SSIM across single-degradation datasets, all-in-one benchmarks and composite-degradation datasets.

**Questions:**

- Could the authors provide more insight into the design rationale behind the different operations? Specifically, what guides the choice between matrix multiplication (Eq. 4), element-wise multiplication (Eq. 5), and convolution + addition (Eq. 6)? Are these design decisions intended to reflect distinct biological processes? If that is the case, what do they reflect, respectively?
- Some components (e.g., the dual-domain loss, the different types of feature interaction in BioModule) might play a foundational role in enabling the rest of the architecture to function effectively.  Have the authors investigated how the performance of P2F and F2P changes when these components are ablated?
- I am also interested in the root cause for the all-in-one image restoration capability of BioIR (without explicit degradation priors). The authors hypothesize that the peripheral branch implicitly captures degradation priors by extracting large-field signals (Lines 263–265). Unfortunately, this lack supporting experimental results. Is there an ablation study that follows the setting in Table 4 and reveals the core design that unlocks the strong all-in-one capability? Or is such capability already presented in the baseline model?
- Table 8(c) suggests that larger receptive fields benefit not only the P2F (which aligns with its contextual role) but also the F2P module, which is intended to capture fine-grained details. This seems counterintuitive. Could the authors explain why F2P benefits from larger kernels, and whether this deviates from the biological analogy?

A clearer clarification (would be better if with experimental analysis and visualization) on the connection between the biology background and the designed BioModule, as well as a thorough ablation study on the all-in-one image restoration capability, would help enhance the impact of this work and increase the rating.

**Ethical Concerns:**

["NO or VERY MINOR ethics concerns only"]

**Final Justification:**

Most of my concerns are addressed.

**Limitations:**

The paper addresses standard limitations, but the discussion around potential negative societal impacts (e.g., misuse of restoration models for manipulating images in sensitive contexts) is not explicitly covered and could be expanded. Additionally, a more explicit treatment of failure cases, particularly on unexpected or compound degradations, would be beneficial.

**Paper Formatting Concerns:**

No significant formatting issues noted.

**Quality:**

3

**Strengths And Weaknesses:**

The work is technically solid: the operators are carefully engineered, ablations are thorough, and experimental coverage is broad (single, unified and composite degradations). Unlike prior works that specialize in a single setting, BioIR demonstrates strong performance on all three image restoration paradigms, despite not relying on explicit degradation priors. BioIR is also shown to be lightweight and fast. Visualizations and architectural figures are clear and helpful in conveying the design intuitions.

That said, several aspects could be improved.

1. while the biological analogy provides a good narrative, the exact correspondence between the proposed modules and actual biological mechanisms feels largely functional/metaphorical rather than mechanistic. The observed performance gains could plausibly be attributed to other architectural features (which are also not employed by previous methods, such as richer cross-scale fusion and the dual-domain loss function), rather than specifically to bio-inspiration. Also, the math around the proposed modules requires more intuition (see "Questions").
2. the writing suffers from some misuse of terminology and inconsistent framing. Note that terms like “generalizability” (Line 32), “universality” (Line 42), and “versatility” (Line 78) have nuanced difference, and their interchanged use here blurs important distinctions. "Universality" focuses more on covering a wide range of existing scenarios, yet "generalizability" implies the adaptability to any unseen scenario (which can be totally artificial). These terms should be applied more precisely. Additionally, the claim of "universal" image restoration is somewhat overstated, as typical tasks like super-resolution and JPEG deblocking are not addressed.
3. Apart from these, there are a few minor weaknesses (mainly on the presentation) that detract from the paper:
   - The orange bounding box in the left part of Figure 1 is confusing. What does it mean?
   - Although NeurIPS does not limit the number of references, there are too many references for introducing the background and related work (CNN-, Transformer-, Mamba-like models are already well-known in 2025). Many citations are only superficially discussed. Please consider providing a more focused and selective review, so the readers feel less confused when tryinig to track specific claims in the paper.
   - In Lines 145--146, The order of $\mathcal{F}$ and $\mathcal{B}$ are reversed ($\mathcal{F}$ should point to the FFN).
   - In Tables 1--3, I suggest removing the publication venues/subscripts for each baseline method, as they clutter the tables without adding meaningful information.
   - The use of gray highlighting is inconsistent across tables. In Tables 1, 2, 5, and 6, gray boxes are used to highlight the results from BioIR models. But in Tables 3, 4 and 7, the boxes are marked gray to convey different meanings (some boxes even take no color, see the last column of Table 3). In Table 8, the gray color seems to indicate the best option in the ablation study. This will confuse readers.
   - Given the complexity of the proposed BioModule, an efficiency-performance bar chart (like Figure 2 in the paper: "A ConvNet for the 2020s.") would help distill practical insights more clearly.

---

> ### Author Rebuttal · Authors · 2025-07-31
>
> We sincerely appreciate the reviewer's insightful comments and the encouraging recognition of our work. Below, we address the raised concerns point by point.
>
> $\colorbox{pink}{Q4.1}$  *more explanations  (weakness 1/question 1)*
>
> Please refer to Q3.1.
>
> $\colorbox{pink}{Q4.2}$  *terminology*
>
> We will make the revisions accordingly. To further demonstrate the universality and generalizability of our model, we conduct additional evaluations on domain-specific tasks and previously unseen datasets.
>
> Specifically, we conduct experiments on ultra-high-definition (UHD) image restoration (4K), medical image restoration, and remote sensing image restoration. We compare our model with several recent algorithms within each individual domain. Tables 8 to 12 show that our model extends effectively to these domain-specific image restoration tasks, outperforming existing methods while maintaining high parameter efficiency. For UHD image restoration, we set the number of BioBlocks in our tiny model to [1,1,1,1,1,1,4] for fair comparison.
>
> **Table 8: Results on UHD-Blur for UHD motion deblurring.**
> |Method|PSNR|SSIM|Params/M|
> |-|-|-|-|
> |FFTformer[1]|25.41|0.725|16.6|
> |UHDFormer[2]|28.82|0.844|0.339|
> |UHDDIP[3]|29.51|0.858|0.81|
> |ERR[4]|29.72|0.861|1.131|
> |Ours|30.40|0.875|0.88|
>
> **Table 9: PET image synthesis on PolarStar M660.**
> |Method|PSNR|SSIM|Params/M|
> |-|-|-|-|
> |DRMC[5]|36.00|0.935|0.62|
> |Restore-RWKV-light[6]|36.96|0.943|1.16|
> |Ours-T|37.17|0.946|1.32|
>
> **Table 10: CT image denoising on AAPM.**
> |Method|PSNR|SSIM|Params/M|
> |-|-|-|-|
> |DenoMamba[7]|33.53|0.915|112.62|
> |Restore-RWKV-light[6]|33.64|0.918|1.16|
> |Ours-T|33.74|0.919|1.32|
>
> **Table 11: MRI image super-resolution on IXI.**
> |Method|PSNR|SSIM|Params/M|
> |-|-|-|-|
> |F-UNet[8]|31.26|0.931|32.12|
> |Restore-RWKV-light[6]|31.36|0.931|1.16|
> |Ours-T|31.50|0.933|1.32|
>
> **Table 12: Remote sensing dehazing results (PSNR/SSIM) on SateHaze1k. The dataset contains three levels of haze, and separate models are trained for each level.**
> |Method|Thin|Moderate|Thick|Params/M|
> |-|-|-|-|-|
> |Trinity[9]|22.65/0.896|24.73/0.934|20.57/0.824|-|
> |FocalNet[10]|24.16/0.916|25.99/0.947|21.69/0.847|3.74|
> |FMambaIR[11]|24.58/0.912|25.83/0.939|22.65/0.850|4.20|
> |Ours-T|24.62/0.922|27.46/0.938|22.98/0.850|1.32|
>
> In addition, to further demonstrate the generalizability of our model, we apply the model pre-trained in the five-task all-in-one setting to previously unseen tasks, including defocus deblurring and desnowing. Tab. 6 shows that our model outperforms strong all-in-one methods across most evaluation metrics.
>
> Regarding the term *universal image restoration*, we conduct experiments across three major tracks of image restoration tasks. Compared to MoCE-IR (*any image restoration*), our evaluation additionally covers nine datasets for single-degradation settings and an extra composite degradation benchmark. Furthermore, in contrast to works on *universal image restoration* such as [12], we include both single-task and composite degradation scenarios. That said, we will consider revising our manuscript to avoid overstating our contributions.
>
> $\colorbox{pink}{Q4.3.1}$  *orange bounding box*
>
> The bounding box surrounding Instances A to D indicates that our model can be applied to multiple image restoration tasks using different model copies, highlighting the universality of our approach.
>
> $\colorbox{pink}{Q4.3.2}$  *references*
>
> In the Introduction and Related Work sections, we provide a broad discussion of prior works related to universality and efficiency, interwoven with concrete examples of relevant methods. Given the broad applicability of our method across multiple scenarios, we review related works in these contexts to highlight the wide-ranging utility of our approach. However, we will consider providing a more focused and selective review in a future revision.
>
> $\colorbox{pink}{Q4.3.3}$  *typo*
>
> We will fix them.
>
> $\colorbox{pink}{Q4.3.4}$  *venues*
>
> We included the venues to highlight the competitive performance of our approach relative to recent methods, particularly those from less familiar tracks. However, as suggested, we will remove them in the revised version.
>
> $\colorbox{pink}{Q4.3.5}$  *gray color*
>
> As suggested, we will either use different colors to distinguish these functions or remove color usage entirely.
>
> $\colorbox{pink}{Q4.3.6}$  *bar chart*
>
> We greatly appreciate your helpful suggestion and will consider adopting this format in a future version.
>
> $\colorbox{pink}{Q4.4}$  *more ablation studies for modules (question 2)*
>
> It is worth noting that our training configurations strictly follow those of previous methods, without introducing any additional tricks. For instance, the dual-domain loss function has been widely adopted in the image restoration field in recent years, across both single-degradation and all-in-one settings, as evidenced by references [1,26,27,29,33,46,47] in the paper. As suggested, removing the FFT loss leads to a 0.74 dB decrease in PSNR (from 33.67 dB to 32.93 dB).
>
> In P2F, the strategy for generating attention weights is crucial. In our model, these weights are derived from a pixel-to-region affinity map. To evaluate the impact of this design, we conducted an additional experiment where the attention weights were instead generated by directly adjusting the channel dimensions of the peripheral features via a convolution. This alternative approach yielded a PSNR of only 34.15 dB, notably lower than the 36.64 dB achieved with our original design. This result underscores the importance of the attention weight generation strategy in the effectiveness of P2F.
>
> In F2P, we adopt a static-to-dynamic strategy. As shown in Table 11 of the paper, we have conducted an ablation study by removing the static part. Here, we further evaluate the effect of removing the dynamic component instead. This variant yields a PSNR that is 1.27 dB lower than our model (36.96 dB), highlighting the critical contribution of the dynamic component to overall performance.
>
> For ablations for the recalibration mechanism, please refer to Tab. 5.
>
> $\colorbox{pink}{Q4.5}$  *ablation studies in all-in-one*
>
> It is worth mentioning that our baseline is built upon Restormer, which is also used as the baseline model in previous all-in-one methods, including AdaIR, Art-PromptIR, and PromptIR. In addition, our training configuration closely follows prior works such as PromptIR, AdaIR, and MoCE-IR, without incorporating any additional tricks. We will release the code and pre-trained models.
>
> As suggested, we conduct ablation studies under the three-task setting due to the limited time available during the rebuttal period. Tab. 13 presents the averaged results obtained by removing each proposed module from the full model. Removing F2P results in a 0.15 dB drop in PSNR, whereas removing P2F, which transfers large-scale contextual information to local regions, leads to a more pronounced performance degradation.
>
> **Table 13: Ablation studies under the three-task all-in-one setting.**
> |Method|PSNR|SSIM|
> |-|-|-|
> |w/o P2F|32.51|0.919|
> |w/o F2P|32.72|0.920|
> |Full | 32.87 | 0.921|
>
> Despite not employing any explicit degradation-aware priors, our model achieves competitive results in all-in-one settings. It does, however, incorporate dynamic mechanisms, such as affinity map generation in P2F and attention weight generation in F2P. Overall, our model serves as a strong baseline and can be further enhanced by integrating explicit priors.
>
> $\colorbox{pink}{Q4.6}$  *large kernel in F2P*
>
> We leverage the proposed modules to emulate foveal and peripheral vision, extracting fine-grained spatial details and broader contextual information, respectively. Both modules benefit from larger convolutional kernels. We believe this phenomenon aligns with the human visual system: it would be advantageous if the fovea could perceive larger fine-detail regions. However, compared to the rod cells in peripheral vision, cone cells in foveal vision require more ganglion cell connections and incur greater computational overhead in the brain. As a result, increasing the number of cone cells would lead to significantly higher processing complexity than that of rod cells. Moreover, since "large" and "small" are relative concepts, we conduct ablation studies using different kernel sizes in P2F and F2P to determine an appropriate configuration that balances accuracy and computational cost.
>
> $\colorbox{pink}{Q4.7}$  *limitation*
>
> We will expand the discussion of our limitations to include additional cases of potential negative societal impacts.
>
> >Reference
>
> [1] Efficient frequency domain-based transformers for high-quality image deblurring, CVPR23.
>
> [2] Correlation matching transformation transformers for uhd image restoration, AAAI24.
>
> [3] Ultra-high-definition restoration: New benchmarks and a dual interaction prior-driven solution, arxiv24.
>
> [4] From zero to detail: Deconstructing ultra-high-definition image restoration from progressive spectral perspective, CVPR25.
>
> [5] Drmc: A generalist model with dynamic routing for multi-center pet image synthesis, MICCAI, 2023.
>
> [6] Restore-rwkv: Efficient and effective medical image restoration with rwkv, IEEE Journal of Biomedical and Health Informatics, 2025.
>
> [7] DenoMamba: A fused state-space model for low-dose CT denoising, arxiv24.
>
> [8] Fourier Convolution Block with global receptive field for MRI reconstruction, Medical Image Analysis, 2025.
>
> [9] Trinity-net: Gradient-guided swin transformer-based remote sensing image dehazing and beyond, IEEE Transactions on Geoscience and Remote Sensing, 2023.
>
> [10] Focal network for image restoration, ICCV23.
>
> [11] FMambaIR: A hybrid state space model and frequency domain for image restoration, IEEE Transactions on Geoscience and Remote Sensing, 2025.
>
> [12] ProRes: Exploring Degradation-aware Visual Prompt for Universal Image Restoration, arxiv23.
>
> [13] HorNet: Efficient High-Order Spatial Interactions with Recursive Gated Convolutions, NeurIPS22.

---

> > ### Comment · Reviewer_jF93 · 2025-08-04
> >
> > Thank you for the detailed feedback.
> >
> > To be clear, for "all-in-one image restoration capability", I was wondering if there's an experiment that shows:
> >
> > | Variant | PSNR (across tasks) | SSIM (across tasks) |
> > | :--- | :---: | :---: |
> > | Baseline w/o module A | (significant decrease)  | (significant decrease) |
> > | Baseline w/ module A | (good) | (good) |
> >
> > So we know exactly what triggers all-in-one image restoration capability of BioIR without explicit degradation priors; Dropping this module leads to performance degradation (like mutual interference of different tasks). Table 13 seems to show that dropping either P2F or F2P does not affect all-in-one restoration much (weakened but not totally lost).
> >
> > Did the authors observe such results? Or, is it true that the BioIR backbone does not have such mutual-interference issue and generally can deal with multiple tasks when we simply feed data from different tasks for training?

---

> ### Author Response · Authors · 2025-08-04
> **Responce to Reviewer jF93**
>
> We appreciate your insightful discussion and would like to respond to your question from two perspectives.
>
> **(a)** Because the performance differences among existing state-of-the-art all-in-one algorithms are marginal, we believe the performance drop observed in the ablation studies of Table 13 is substantial. To provide a clearer comparison, we have redrawn the table to include previous competing methods. As shown in Table 14, removing our proposed modules results in performance inferior to that of previous methods, underscoring the significance of the proposed components.
>
> **Table 14**
> |Method|PSNR|
> |-|-|
> |BioIR w/o P2F|32.51|
> |BioIR w/o F2P |32.72|
> |UniProcessor [1]   |  32.70  |
> |AdaIR [2]  |32.69 |
> |MoCE-IR [3]  |32.73 |
> |BioIR | 32.87 |
>
> **(b)** *Can methods without explicit priors outperform those that rely on them?* We believe the answer is yes. Supporting evidence can be found in the results presented in Ref. [4] (Table II). For the reviewer’s convenience, we reproduce some of these results in Table 15. The top two methods listed are general algorithms, while the bottom two are all-in-one approaches. Two conclusions can be drawn from the table: (*i*) General image restoration methods can achieve comparable or even superior performance to all-in-one approaches, while using fewer parameters; and (*ii*) architecture design plays a crucial role in all-in-one performance, *e.g.*, FSNet achieves competitive results with significantly fewer parameters.
>
> **Table 15 Excerpt from Table II in Ref. [4].**
> |Method|PSNR|#Params|
> |-|-|-|
> |FSNet  | 31.42 | 13.28M |
> |MambaIR | 31.51| 26.78M |
> |ProRes | 30.38 | 370.63M |
> |NDR | 31.51 | 28.40M |
>
> Given the above analysis, we believe that strong performance on all-in-one tasks can be achieved through effective architecture design that enhances model capacity. While our model does not rely on explicit priors, it incorporates dynamic mechanisms that implicitly capture differences across inputs, such as the affinity map–based weight generation in P2F, and the gated operation and dynamic convolution in F2P.
>
> Overall, our model demonstrates that with proper architectural design, even without explicit priors, it can achieve state-of-the-art performance on all-in-one tasks. We also believe that introducing explicit priors or a dynamic learning strategy to balance different tasks could further improve performance. In this regard, our model has strong potential to serve as a baseline for future research. We will release the code and pre-trained models to support reproducibility and contribute to the community.
>
> >Reference
>
> [1] Uniprocessor: a text-induced unified low-level image processor, ECCV24.
>
> [2] Adair: Adaptive all-in-one image restoration via frequency mining and modulation, ICLR25.
>
> [3] Complexity experts are task-discriminative learners for any image restoration, CVPR25.
>
> [4] Perceive-ir: Learning to perceive degradation better for all-in-one image restoration, TIP25.

---

> > ### Comment · Reviewer_jF93 · 2025-08-04
> >
> > I appreciate the discussion.
> >
> > Let me be a bit more clearer, Table 15 seems to be comparing specialized models and all-in-one models on each single task, meaning that the non-universal models are trained and tested with only data from that task (so there are separate weights for each different task). In this case, it is totally reasonable to see a universal models lag slightly behind the specialized ones.
> >
> > What I was curious about: (1) training a specialized baseline with mixed data from multiple tasks; (2) training the same baseline plus some module (which turns the specialized baseline to an all-in-one model) with the same mixed data from multiple tasks; (3) test the models under each task, which i think aligns with your experiment setting in Line 234.
> >
> > | Variant | Arch. Type | Training | expected PSNR (across tasks) | expected SSIM (across tasks) |
> > | :--- | :---: | :---: |:---: | :---: |
> > | Baseline w/o module A | specialized | trained/tested on a single task | (good)  | (good) |
> > | Baseline w/o module A | specialized | trained w/ mixed data from multiple tasks | (significant decrease)  | (significant decrease) |
> > | Baseline w/ module A | all-in-one | trained/tested on a single task |  (good) | (good) |
> > | Baseline w/ module A | all-in-one | trained w/ mixed data from multiple tasks |  (good) | (good) |
> >
> > If I miss such ablation in the paper or misunderstand any points, feel free to point it out.

---

> ### Author Response · Authors · 2025-08-04
> **Response to Reviewer jF93**
>
> We appreciate your thoughtful discussion and the table provided.
>
> We believe there may have been a misunderstanding regarding the results presented in Table 15. We apologize for any confusion caused.
>
> To clarify, the results in Table 15 (averaged across tasks) were obtained by retraining general algorithms (FSNet and MambaIR, not all-in-one models) without any architectural modifications, under the same three-task setting, as described in Ref. [4] (specifically, in the caption of their Table II). In this setting, the models are trained on a mixed dataset comprising data from multiple tasks and evaluated on the corresponding individual test sets.
>
> It is important to note that the three-task (and five-task) all-in-one setting is a standard evaluation protocol used in works such as MoCE-IR [3], AdaIR [2], Ref. [4], and ours.
>
> Table 15 shows that even general algorithms, without any architectural modifications, can outperform some (though not all) methods specifically designed for all-in-one image restoration, when trained under the same all-in-one setting. This suggests that with an appropriate architecture, strong performance on all-in-one tasks can be achieved even without relying on explicit priors.
>
> This observation supports the feasibility of our model achieving competitive performance in all-in-one scenarios. In fact, our results confirm this, demonstrating that strong performance can be achieved without relying on explicit priors.
>
> Please note that all recent discussions we have provided are unrelated to the performance on a single task (the third row in your table, trained/tested on a single task), but instead focus on the impact of architectural changes on all-in-one performance.

---

> > ### Comment · Reviewer_jF93 · 2025-08-04
> >
> > I have no further comments, although i expect such a table as listed above to find out the core design of BioIR that unlocks all-in-one capability (i am not skeptical for the performance of BioIR, but care more about the ablation). The authors generally made a good rebuttal and I am raising my score.

---

> > > ### Author Response · Authors · 2025-08-04
> > > **Response to Reviewer jF93**
> > >
> > > We thank the reviewer for the constructive comments and the encouraging remark regarding a score increase. We would sincerely appreciate it if this update could be reflected in the review system. Thank you again for your time.

---

### Official Review · Reviewer_AN93 · 2025-06-30

**Clarity:** 3
**Significance:** 2
**Originality:** 3
**Rating:** 5
**Confidence:** 4

**Summary:**

This paper presents BioIR, a novel bio-inspired image restoration framework that emulates the functional interplay between foveal and peripheral pathways in the human visual system. The proposed architecture incorporates two key modules: the Peripheral-to-Foveal (P2F) module, which leverages pixel-to-region affinity to propagate large-field contextual signals, and the Foveal-to-Peripheral (F2P) module, which employs a static-to-dynamic integration strategy to enhance fine-grained spatial details. Extensive experiments across single-degradation, all-in-one, and composite degradation settings demonstrate that BioIR achieves state-of-the-art performance while maintaining high computational efficiency.

**Questions:**

1. Explain why P2F and F2P are bio-inspired. The interpretability of feature interactions is not sure.
2. The visual comparisons are mainly from synthetic degraded images. Real world captured images are preferred to show the generalization of BioIR.

**Ethical Concerns:**

["NO or VERY MINOR ethics concerns only"]

**Limitations:**

See Questions.

**Quality:**

3

**Strengths And Weaknesses:**

1. BioIR outperforms existing methods on multiple benchmarks (e.g., PSNR=31.69  on SOTS-Indoor for dehazing and  PSNR=23.29 on LOL for low-light enhancement) with less parameters. The SOTA methods are the latest with many visual comparisons provided.
2. Peripheral-to-Foveal (P2F) and Foveal-to-Peripheral (F2P) modules in BioModule is a contribution of this paper, which synergistically combine global context and local detail refinement. However, it should be explained clearly why P2F and F2P are bio-inspired. In the network architecture, it can only be observed that P2F and F2P are interacted dual-stream.

---

> ### Author Rebuttal · Authors · 2025-07-31
>
> We sincerely thank Reviewer AN93 for recognizing the strengths of our method and for the constructive feedback. We address the raised issues point by point below.
>
> $\colorbox{pink}{Q3.1}$ *more explanations*
>
> We explain our bio-inspired design on two levels. The first level mimics the perception process of the human visual system through a three-stage mechanism, as illustrated in Figure 2. The second level involves the design of specific modules inspired by key functional characteristics of the visual system, rather than a one-to-one mapping to biological structures. The following parts describe how these functional mechanisms are emulated in our model design.
>
> **(a)** Unlike the human visual system, which can focus on only one position at a time, deep learning models are capable of learning patterns across the entire feature space. Accordingly, we learn pixel-wise affinity maps and attention weights in the P2F and F2P modules, respectively.
>
> **(b)** In the retina, cone cells are concentrated around the fovea to capture fine-grained details, while rod cells cover a much larger area to support contextual peripheral vision. Cone cells, in contrast to rod cells, require more ganglion cell connections and lead to increased computational overhead in the brain. As a result, increasing the size of the foveal region or reducing the peripheral region leads to higher computational complexity.
>
> To mimic this variation, we apply spatial pooling to extract contextual signals in the P2F module and perform matrix multiplication after pooling. This design ensures that the complexity is proportional to $S^2$ (downsampled spatial size), which is inversely related to the size of the peripheral region. Hence, the complexity of affinity generation in P2F decreases as the peripheral region increases. In contrast, in F2P, we directly generate attention weights after a depth-wise convolution, making the computational cost proportional to the kernel size. Tables 8(b) and 8(c) illustrate how the complexity varies with scope/kernel size in both modules.
>
> **(c)** To simulate the bidirectional interaction between foveal and peripheral vision, we employ dynamic convolution in both modules, enabling content-aware and input-adaptive visual integration. This design is inspired by the adaptability of biological receptive fields, which are modulated by external stimuli.
>
> **(d)** In F2P, since foveal vision focuses on extracting fine-grained local features, we integrate two complementary mechanisms: dynamic convolution and element-wise multiplication. Dynamic convolution enables content-dependent, spatially adaptive transformations, analogous to high-level semantic modulation in the human visual system. In contrast, element-wise multiplication serves as a low-level gating mechanism or gain control. By summing the outputs of both pathways, the model is able to fully exploit the valuable fine-grained information provided by the foveal stream.
>
>
> $\colorbox{pink}{Q3.2}$ *generalization evaluation*
>
> Thank you for the helpful comment. We provided the visual comparisons for real-world low-light deblurring in Figure 10 of the paper. However, due to the conference policy, we are currently not permitted to update the manuscript or include an external link to provide additional real-world visual results. As an alternative way to demonstrate the generalization capability of our model, we apply the model pre-trained under the five-task all-in-one setting to previously unseen tasks, including defocus deblurring and desnowing. As shown in Tab. 7, our model outperforms recent strong all-in-one approaches on most metrics while maintaining high parameter efficiency.
>
> **Table 7: PSNR and SSIM scores for out-of-distribution evaluation.**
> |Method | DPDD (Defocus deblurring) | SRRS (Desnowing) | #Params |
> |-| -| - | - |
> |AdaIR [1] |15.49/0.601| 21.86/0.845| 29M|
> |MoCE-IR [2] |15.58/0.598 |21.82/0.838 | 25M |
> |Ours |17.80/0.629|21.93/0.841 | 16M |
>
> >Reference
>
> [1] Adair: Adaptive all-in-one image restoration via frequency mining and modulation, ICLR25.
>
> [2] Complexity experts are task-discriminative learners for any image restoration, CVPR25.

---

### Official Review · Reviewer_rjfQ · 2025-07-01

**Clarity:** 4
**Significance:** 3
**Originality:** 2
**Rating:** 5
**Confidence:** 3

**Summary:**

The paper introduces BioIR, an efficient and universal image restoration framework inspired by the human visual system. Unlike prior task-specific or degradation-aware methods, BioIR generalizes across single-degradation, all-in-one, and composite degradation settings, while maintaining high computational efficiency.

BioIR emulates human vision through two novel modules:

Peripheral-to-Foveal (P2F): Injects large-field contextual signals into local (foveal) regions using pixel-to-region affinity.

Foveal-to-Peripheral (F2P): Propagates fine-grained spatial details outward using a two-stage static-to-dynamic integration strategy.

The network architecture follows a U-shaped design with BioModules replacing self-attention in Transformer-style blocks. This bio-inspired design promotes dynamic interaction between global and local features and is shown to boost both accuracy and efficiency.

Through comprehensive experiments, BioIR achieves:

State-of-the-art performance on 9 datasets across 4 single-degradation tasks (e.g., dehazing, deraining, desnowing, low-light enhancement).

Superior results in 3-task and 5-task all-in-one restoration without explicit degradation priors.

Leading results on composite degradation datasets (e.g., CDD11, LOLBlur), outperforming models with more parameters.

An ablation study confirms the contribution of P2F, F2P, and the recalibration mechanism, and runtime tests show fast inference on high-resolution images.

**Questions:**

The proposed modules are motivated by biological inspiration. However, in practice, the implementation mainly involves computing an affinity matrix, and there is neither qualitative nor quantitative evidence to support whether the modules function as intended. Could the authors provide visualizations or empirical results that demonstrate how the modules behave in line with the claimed design?

Moreover, there appears to be no theoretical analysis explaining why the proposed approach performs better than prior methods. Are there any theoretical insights or analyses that can justify the effectiveness of the design?

Lastly, it is still unclear how the proposed modules, described as bio-inspired in the introduction, are truly analogous to biological visual processing. Why did the authors choose to frame the design as "bio-inspired"?

**Ethical Concerns:**

["NO or VERY MINOR ethics concerns only"]

**Final Justification:**

The authors have sufficiently addressed the concerns and limitations I raised, and I am updating my score to Accept based on their responses.

**Limitations:**

Conceptual–Implementation Gap:
Although the modules are claimed to be biologically inspired, the actual implementation largely reduces to affinity-based feature interactions and element-wise operations. There is a lack of direct evidence—either qualitative visualizations or quantitative evaluations—that these mechanisms faithfully reflect biological perceptual processes.

Lack of Theoretical Analysis:
The paper does not provide a theoretical explanation for the observed performance gains. Without insights into the model's expressivity, inductive bias, or optimization advantages, it is difficult to understand why the proposed modules outperform conventional designs.

Ambiguity of the "Bio-Inspired" Framing:
The rationale for describing the architecture as "bio-inspired" remains vague. The analogy to foveal and peripheral vision is conceptually interesting but not rigorously defined or validated. The current design might better be described as structurally inspired rather than biologically faithful.

**Quality:**

3

**Strengths And Weaknesses:**

## Strengths
Innovative Bio-Inspired Architecture: The proposed P2F (Peripheral-to-Foveal) and F2P (Foveal-to-Peripheral) modules draw inspiration from the functional interactions in the human visual system. This biologically motivated design introduces a novel perspective on contextual integration and detail propagation in image restoration.

Strong Generalization Across Tasks: BioIR demonstrates remarkable versatility by achieving state-of-the-art performance across three representative image restoration settings: single degradation, all-in-one, and composite degradation. This makes the model suitable for diverse real-world scenarios.

High Computational Efficiency: Despite strong performance, the model is highly efficient. The tiny version (BioIR-T) achieves competitive results, significantly reducing inference time and resource requirements compared to existing baselines.

Comprehensive Experimental Validation: The model is thoroughly evaluated on 9 datasets covering various degradation types and is benchmarked against strong baselines. Both quantitative and qualitative results support the effectiveness of the proposed method.

## Weaknesses
Lack of Theoretical Justification: While the bio-inspired modules are intuitively motivated, the paper does not provide rigorous theoretical analysis to justify why the proposed interactions between foveal and peripheral features are optimal or fundamentally effective.

Simple Recalibration Mechanism: The recalibration step between P2F and F2P streams relies on element-wise multiplication, which may be overly simplistic. More sophisticated fusion strategies could potentially yield better results.

Limited Adaptivity to Unknown Degradations: The model lacks explicit degradation-type prediction or task-adaptive mechanisms. In contrast, some prior works incorporate degradation-aware priors or task-conditioned prompts.

Biological Analogy Not Empirically Validated: While the architecture is inspired by the human visual system, the paper does not provide empirical or neuroscientific evidence to support the analogy beyond intuitive similarity.

---

> ### Author Rebuttal · Authors · 2025-07-31
>
> We sincerely thank Reviewer rjfQ for the valuable time and thoughtful feedback. We respond to your comments point by point below.
>
> $\colorbox{pink}{Q2.1}$  *more explanations for the modules (question 1)*
>
> Inspired by the hierarchical processing of the human visual system, we loosely organize our model into three functional stages: encoding, interaction, and recalibration [1]. While these stages do not strictly map to anatomical counterparts, they reflect a coarse functional analogy: early encoding extracts spatial features, intermediate interaction enables complementary integration of foveal and peripheral information, and final recalibration adjusts feature responses based on their mutual relevance. To implement these concepts, we utilize fundamental deep learning components. Specifically, we employ downsampling and depth-wise convolution to extract large-scale contextual signals and refine local features, respectively. Next, we introduce dynamic mechanisms to integrate the learned representations, inspired by the adaptability of biological receptive fields modulated by stimuli. Finally, we apply a simple element-wise multiplication for feature recalibration, leveraging its simplicity, parameter efficiency, and capacity to model non-linear dependencies between the two streams.
>
> On the other hand, we acknowledge inherent differences between the biological visual system and deep learning architectures. For instance, the human visual system typically focuses on a single position at a time, whereas deep models can learn patterns across the entire feature space simultaneously. To reflect this distinction, our model learns a pixel-wise affinity map in the P2F module and a pixel-wise attention weight in the F2P module. Furthermore, we visualize the features after spatial pooling in P2F and depth-wise convolution in F2P; the former captures large-scale semantic information, while the latter retains fine-grained spatial details. Due to conference policy, we are unable to update the manuscript or provide an external link at this stage, but we will include these visualizations in a future version of the paper.
>
> Please refer to our response to Q3.1 for further explanation of the module design.
>
> Additionally, we explicitly acknowledge in the Limitation section that human visual perception is highly complex and remains an open research area. Our proposed architecture represents a preliminary and conceptual attempt to mimic certain aspects of this process. We agree that a more rigorous theoretical analysis would further strengthen our work, and we consider this an important direction for future research.
>
> $\colorbox{pink}{Q2.2}$ *justification for results (question 2)*
>
> While we do not claim theoretical optimality, we believe that the model’s effectiveness can be attributed to its architectural bias: (1) a vision system-inspired three-stage mechanism forms the foundation of our design; (2) the interaction and recalibration stages serve as efficient information fusion modules, effectively complementing feature representations and selectively enhancing those that are most relevant to the task; (3) the dynamic affinity-based integration introduces adaptability that enhances generalization across varying degradations, and (4) the overall design strikes a deliberate balance between effectiveness and efficiency. For example, as shown in Tab. 5, a more advanced recalibration module yields slightly better performance but introduces significantly higher computational overhead, exceeding that of our final model, despite its inferior overall score. Moreover, the effectiveness of our approach is supported by extensive ablation studies on architectural choices and hyperparameter settings.
>
> Overall, our model incorporates several key components commonly found in advanced architectures, including multi-scale representation learning, dynamic feature integration via affinity map generation, and efficient interactions between foveal and peripheral representations. Together, these elements contribute to the model’s strong performance across three major image restoration tasks. We also emphasize that our training configurations strictly follow prior works to ensure fair comparisons. To support reproducibility and future research, we will release the code and pre-trained models.
>
> $\colorbox{pink}{Q2.3}$ *bio-inspired? (question 3)*
>
> Our use of the term *bio-inspired* is intended to reflect inspiration from the mechanisms observed in the human visual system, particularly the cooperative roles and interactions between high-acuity (foveal) and broad-context (peripheral) processing, rather than to imply a strict biological or anatomical replication. The proposed modules aim to abstract key ideas such as hierarchical encoding, adaptive integration, and efficient modulation, which are broadly aligned with how biological vision balances efficiency and precision. While our current design does not model detailed physiological processes, we believe the bio-inspired terminology remains appropriate in the context of mechanism-level inspiration.
>
> Nonetheless, we understand the importance of precision in terminology, and we will revise the manuscript to better clarify this framing and avoid potential misinterpretation.
>
> $\colorbox{pink}{Q2.4}$ *sophisticated recalibration mechanism*
>
> Our original intention behind the recalibration design was to recalibrate the two branches while maintaining high efficiency. We acknowledge that more sophisticated strategies may further improve performance. As you suggest, we first replace our recalibration method with simple alternatives, including addition and channel concatenation followed by a convolutional layer (to adjust channel dimensions). However, neither alternative outperforms our design in terms of accuracy or computational efficiency. We further explore more sophisticated variants, such as channel-wise cross-attention and recursive gated convolutions (g$^n$conv) [2]. While these achieve better performance, they incur significantly higher computational overhead, which conflicts with our goal of developing an efficient and effective module. This trade-off is further highlighted by our final model outperforming both variants in terms of PSNR, FLOPs, and parameter count.
>
> **Table 5: Alternatives of the recalibration operation.**
> |Method|PSNR|FLOPs|#Params|
> |-|-|-|-|
> |Mul.(Ours)  |  33.67 | 14.34 |1.23M|
> |Add  | 33.28 | 14.34 | 1.23M|
> |Concat+CNN | 33.65 | 15.97 | 1.39M |
> |Cross-att. | 35.98| 16.76 | 1.55M|
> |g$^n$conv [2] | 35.47 | 18.27 | 1.60M|
> |Our final model | 38.26 | 16.65| 1.32M|
>
> $\colorbox{pink}{Q2.5}$ *unknown degradations*
>
> As suggested, we evaluate the pre-trained model in the five-task all-in-one setting on out-of-distribution tasks, including defocus deblurring and desnowing. We compare our results with AdaIR (ICLR 2025), which explicitly incorporates degradation-aware frequency priors, and the MoE-based MoCE-IR (CVPR 2025). As shown in Tab. 6, our model outperforms these strong baselines across most evaluation metrics while maintaining superior parameter efficiency. Although AdaIR and MoCE-IR use sub-networks to extract degradation-related priors for distinguishing degradation types, these sub-networks still struggle to generalize to unseen datasets. In contrast, our model incorporates dynamic mechanisms, such as affinity generation, that enhance its representational capacity in the all-in-one setting. Therefore, our model demonstrates strong potential as a baseline that could further benefit from incorporating degradation-aware priors to achieve even better performance.
>
> **Table 6: PSNR and SSIM scores for out-of-distribution evaluation.**
> |Method|DPDD (defocus deblurring)|SRRS (desnowing)|#Params|
> |-|-|-|-|
> |AdaIR [3]|15.49/0.601 | 21.86/0.845| 29M|
> |MoCE-IR [4] |15.58/0.598 |21.82/0.838 | 25M |
> |Ours | 17.80/0.629|21.93/0.841 | 16M|
>
> >Reference
>
> [1] A review of interactions between peripheral and foveal vision, Journal of vision, 2020.
>
> [2] HorNet: Efficient High-Order Spatial Interactions with Recursive Gated Convolutions, NeurIPS22.
>
> [3] Adair: Adaptive all-in-one image restoration via frequency mining and modulation, ICLR25.
>
> [4] Complexity experts are task-discriminative learners for any image restoration, CVPR25.

---

> > ### Comment · Reviewer_rjfQ · 2025-08-03
> >
> > Thank you for your detailed response to my review. I have carefully examined your revisions and believe that the limitations and weaknesses I previously noted have been appropriately addressed. Accordingly, I will raise my score. I strongly encourage the authors to release the accompanying code prior to publication to enhance the contribution to the research community.

---

> > > ### Comment · Reviewer_rjfQ · 2025-08-05
> > >
> > > With the extended discussion period, I have a follow-up question. Do you plan to publish code that would allow reproduction of all the experiments?

---

> > > > ### Author Response · Authors · 2025-08-05
> > > > **Response to Reviewer rjfQ**
> > > >
> > > > Thank you for your constructive feedback and for increasing your score.
> > > >
> > > > We confirm that we will release our code and ALL of our pre-trained models to fully support reproducibility and facilitate future research.
> > > >
> > > > Once again, thank you for your valuable time and thoughtful comments.

---

### Official Review · Reviewer_imoR · 2025-07-03

**Clarity:** 3
**Significance:** 3
**Originality:** 2
**Rating:** 3
**Confidence:** 5

**Summary:**

The authors present BioIR, an efficient and universal image restoration framework inspired by the human visual system. Specifically,they design two bio-inspired modules, Peripheral-to-Foveal (P2F) and Foveal-to-Peripheral (F2P), to emulate the perceptual processes of human vision, with a particular focus on the functional interplay between foveal and peripheral pathways. BioIR network achieves state-of-the-art performance across three major categories of image restoration tasks, including nine datasets for four single-degradation tasks, two all-in-one task settings, and two composite degradation benchmarks, while maintaining high computational efficiency and fast inference speed.

**Questions:**

Please check the weakness section.
It seems to be typo in table 9 where base and large model seems to have the same channels, blocks.

**Ethical Concerns:**

["NO or VERY MINOR ethics concerns only"]

**Limitations:**

The current major limitation for me is the novelty of the work, which does not seem to be properly addressed in the work.

**Quality:**

2

**Strengths And Weaknesses:**

Strenghts:
The paper is easy to follow.
The paper explores the important aspect of efficiency, and shows comparable results with the state-of-the-art image restoration works on almost all the datasets.

Weaknesses:
1. The main idea of bio-inspired module is not clear, how is it replicating the human visual system and how is it achieving the desired task? The novelty of the work is limited, specifically the 2 modules.
1a) In Figure 3c, after the splitting of channels via DW Conv how can we conclude which is the Peripheral and which is the foveal signal? If the authors would have shown some feature maps or some more ablations on that part may have bit clarified the splitting.
1b) Another point of concernn is the pooling operation used in both the P2F and F2P modules, doesn't it leads to loss of information as well, which is important point to be considered in restoration application like deblurring. What would be the conclusion if they would have alternatively use max pooling?
1c) As mentioned in line 50, that foveal signals are extrapolated to support peripheral representations, how is this extrapolation being done in the proposed module.
2. The authors should include more relevant ablations, atleast on different recalibration options, different configurations of the F2P and P2F modules (not just changing the kernel size as shown in supp, like why dynamic conv works well).

---

> ### Author Rebuttal · Authors · 2025-07-31
>
> We sincerely thank reviewer imoR for the thoughtful comments and constructive suggestions. We address the issues point by point below.
>
> $\colorbox{pink}{Q1.1}$  *bio-inspired modules*:
>
> Please refer to our response to Q3.1 for a detailed explanation of the module design.
>
>
>
> $\colorbox{pink}{Q1.1(a)}$  *splitting operation*:
>
> The features $P$ and $F$ denote the peripheral and foveal signals (or branches), respectively. It is important to clarify that this distinction does not arise from a splitting operation; rather, $P$ and $F$ are conceptually differentiated by the subsequent processing applied to them. For instance, in the P2F branch, spatial average pooling ($C \times H \times W \rightarrow C \times S \times S$) is used to extract peripheral visual information.
>
> In our model, the splitting operation generates two feature sets that are subsequently processed to obtain the peripheral and foveal vision signals. This operation serves two purposes: (1) improving efficiency by applying peripheral and foveal processing to only half of the features, and (2) enhancing feature diversity through interactions between the two halves. To validate this design choice, we perform an ablation study without splitting, where both vision signals originate from the same features. As shown in Tab. 1, these variants perform worse than our original design while also incurring higher computational complexity.
>
> **Table 1: Ablation studies for the splitting operation.**
> |Method|PSNR|FLOPs|#Params|
> |----------|----------|----------|----------|
> |P2F (ours) | 36.64| 15.38 | 1.25M |
> |P2F (w/o splitting)| 35.39 | 16.2| 1.33M |
> |F2P (ours) | 36.96 | 15.61 | 1.29M |
> |F2P (w/o splitting) | 36.92 | 17.67 | 1.43M  |
>
> $\colorbox{pink}{Q1.1(b)}$ *average pooling*:
>
> The use of pooling techniques offers two main advantages: (1) reducing spatial or channel dimensions, which lowers computational overhead and contributes to the model’s efficiency while preserving competitive performance through our architectural design; and (2) enabling large receptive fields via spatial pooling in P2F, effectively mimicking peripheral vision. As shown in Table 8b, decreasing the pooling factor in P2F improves performance but also increases FLOPs.
>
> For deblurring tasks, we conduct an additional experiment on the ultra-high-definition (UHD) motion deblurring dataset, UHD-Blur [2], a challenging 4K benchmark that stresses both performance and efficiency. We train our model following the configuration of the recent CVPR 2025 method, ERR [4]. We set the number of BioBlocks in our tiny model to [1,1,1,1,1,5] for fair comparison. As shown in Tab. 2, our model outperforms the strong competitor ERR by a significant margin of 0.68 dB in PSNR, despite having fewer parameters. This result underscores the effectiveness of our overall design choices.
>
> **Table 2: Results on UHD-Blur for UHD motion deblurring.**
> |Method | PSNR | SSIM | #Params|
> |----------|----------|----------|----------|
> |FFTformer [1] | 25.41| 0.725| 16.6M|
> |UHDFormer [2]|  28.82 |0.844 |0.339M|
> |UHDDIP [3]  |29.51| 0.858 |0.81M|
> |ERR [4] | 29.72 |0.861 |1.131M|
> |Ours |30.40  | 0.875 | 0.88M|
>
> As suggested, we individually replace average pooling with max pooling in FP2 and F2P. As shown in Tab. 3, this replacement results in a performance drop in both modules. This decline is likely due to the fact that average pooling preserves signal patterns within the pooled regions by aggregating information, whereas max pooling retains only the strongest activations. Nonetheless, the models using max pooling still outperform the baseline, highlighting the effectiveness of our overall design.
>
> **Table 3: PSNR scores when replacing average pooling with max pooling in P2F (at two positions) and F2P (at one position).**
> |Method | PSNR |
> |----------|----------|
> Base |33.18|
> |Base+P2F|36.64 (avg.)/35.08 (max)|
> |Base+F2P|36.96 (avg.)/36.59 (max)|
>
> $\colorbox{pink}{Q1.1(c)}$ *extrapolation operation*:
>
> In the biological visual system, the eye can focus on only one location at a time, and foveal signals are often extrapolated to interpret peripheral regions. In contrast, deep learning models are capable of learning dedicated patterns for all regions simultaneously. Therefore, our network does not strictly rely on extrapolating learned foveal signals to peripheral areas. Rather, the perception process of the visual system is introduced primarily for illustrative purposes, and our architecture is designed conceptually to emulate its underlying mechanisms.
>
> To allow the fine-grained spatial signals from foveal vision to benefit peripheral representations, we adopt a two-stage design. In this design, each pixel in the foveal branch is propagated forward either through element-wise multiplication or by generating integration weights based on each pixel. If one were to draw an analogy, the relationship between the pixel-wise weights from the foveal vision and the integrated peripheral region could be loosely interpreted as a form of extrapolation.
>
> $\colorbox{pink}{Q1.2}$ *more ablation studies*:
>
> As suggested, the additional ablation results are presented in Tab. 4. We first replace our recalibration method (multiplication) with simple alternatives, including addition and channel concatenation followed by a convolution layer (to adjust channel dimensions). However, neither alternative outperforms our design in terms of accuracy or computational efficiency. We further explore more complex variants, such as cross-attention and recursive gated convolutions (g$^n$conv) [5]. While these achieve better performance, they incur significantly higher computational overhead, which conflicts with our goal of developing an efficient and effective module. This trade-off is further highlighted by our final model outperforming both variants in terms of PSNR, FLOPs, and parameter count.
>
> **Table 4: Alternatives to the recalibration operation.**
> |Method | PSNR | FLOPs| #Params|
> |----------|----------|----------|----------|
> |Mul.(Ours)  |  33.67 | 14.34 |1.23M |
> |Add  | 33.28 | 14.34 | 1.23M|
> |Concat+CNN | 33.65 |15.97 | 1.39M |
> |Cross-att. | 35.98 | 16.76 | 1.55M |
> |g$^n$conv [5] | 35.47 | 18.27| 1.60M |
> |Our Final | 38.26 | 16.65 | 1.32M|
>
> Regarding the dynamic convolution in P2F, the method for generating attention weights plays a critical role. In our model, these weights are derived from a pixel-to-region affinity map. To assess the impact of this design, we conducted an additional experiment in which the attention weights were generated by directly adjusting the channel count of the peripheral features via a convolution layer. This alternative approach achieved only 34.15 dB in PSNR, which is significantly lower than the 36.64 dB obtained with our original design. This comparison highlights the importance of the attention weight generation strategy in the effectiveness of the dynamic convolution.
>
> $\colorbox{pink}{Q1.3}$ *typo*:
>
> Sorry for the typo. The correct number of BioBlocks in the large model is [6,6,14,14,6,6,4]. We will fix it.
>
> >Reference
>
> [1] Efficient frequency domain-based transformers for high-quality image deblurring, CVPR23.
>
> [2] Correlation matching transformation transformers for uhd image restoration, AAAI24.
>
> [3] Ultra-high-definition restoration: New benchmarks and a dual interaction prior-driven solution, arxiv24.
>
> [4] From zero to detail: Deconstructing ultra-high-definition image restoration from progressive spectral perspective, CVPR25.
>
> [5] HorNet: Efficient High-Order Spatial Interactions with Recursive Gated Convolutions, NeurIPS22.

---

> ### Comment · Area_Chair_b2pf · 2025-08-06
>
> Dear **Reviewer imoR**,
>
> Could you please review the rebuttal and confirm whether the initial questions or concerns have been addressed? Your participation in this author-reviewer discussion would be greatly appreciated. Thank you very much for your time and effort.
>
> Best,
>
> AC

---

### Decision · Program_Chairs · 2025-09-17

**Decision:**

Accept (poster)

**Comment:**

The paper received mostly positive feedback after the authors provided their rebuttal.

Initially, the reviewers expressed several main concerns, including a disconnect between the concept and its implementation, insufficient rigorous justifications, potential information loss due to pooling, unclear details regarding extrapolation, ambiguous terminology, inconsistent framing, and inadequate relevant ablation studies. The authors' rebuttal and additional experiments addressed most of these concerns. After the rebuttal and subsequent discussion, no further issues were raised.

Reviewer imoR, who had initially given a "borderline reject" rating, noted in the follow-up discussion that "the authors made a good rebuttal and addressed most of my concerns," and expressed a willingness to "increase the score from borderline reject to borderline accept."

The area chair concurs with the reviewers' overall evaluations and recommends accepting the paper.